# Cardiomyopathy in Duchenne Muscular Dystrophy and the Potential for Mitochondrial Therapeutics to Improve Treatment Response

**DOI:** 10.3390/cells13141168

**Published:** 2024-07-09

**Authors:** Shivam Gandhi, H. Lee Sweeney, Cora C. Hart, Renzhi Han, Christopher G. R. Perry

**Affiliations:** 1School of Kinesiology and Health Science, Muscle Health Research Centre, York University, Toronto, ON M3J 1P3, Canada; 2Department of Pharmacology and Therapeutics, University of Florida, Gainesville, FL 32610, USA; lsweeney@ufl.edu (H.L.S.); coracricket@ufl.edu (C.C.H.); 3Myology Institute, University of Florida, Gainesville, FL 32610, USA; 4Department of Pediatrics, Herman B Wells Center for Pediatric Research, Indiana University School of Medicine, Indianapolis, IN 46202, USA; rh11@iu.edu

**Keywords:** Duchenne muscular dystrophy, cardiomyopathy, mitochondria, elamipretide, therapy, inflammation, ion dysregulation, sarcolemmal tearing, metabolism, calcium balance, skeletal muscle, gene therapy, antioxidants, bioenergetics, reactive oxygen species

## Abstract

Duchenne muscular dystrophy (DMD) is a progressive neuromuscular disease caused by mutations to the dystrophin gene, resulting in deficiency of dystrophin protein, loss of myofiber integrity in skeletal and cardiac muscle, and eventual cell death and replacement with fibrotic tissue. Pathologic cardiac manifestations occur in nearly every DMD patient, with the development of cardiomyopathy—the leading cause of death—inevitable by adulthood. As early cardiac abnormalities are difficult to detect, timely diagnosis and appropriate treatment modalities remain a challenge. There is no cure for DMD; treatment is aimed at delaying disease progression and alleviating symptoms. A comprehensive understanding of the pathophysiological mechanisms is crucial to the development of targeted treatments. While established hypotheses of underlying mechanisms include sarcolemmal weakening, upregulation of pro-inflammatory cytokines, and perturbed ion homeostasis, mitochondrial dysfunction is thought to be a potential key contributor. Several experimental compounds targeting the skeletal muscle pathology of DMD are in development, but the effects of such agents on cardiac function remain unclear. The synergistic integration of small molecule- and gene-target-based drugs with metabolic-, immune-, or ion balance-enhancing compounds into a combinatorial therapy offers potential for treating dystrophin deficiency-induced cardiomyopathy, making it crucial to understand the underlying mechanisms driving the disorder.

## 1. An Overview of the Cellular, Physiological, and Clinical Manifestations of DMD

### 1.1. Introduction

Duchenne muscular dystrophy (DMD) is a devastating, progressive neuromuscular disorder caused by X-linked recessive mutations to the dystrophin gene (*DMD*) located on chromosome Xp21 [1,2]. The mutation results in a deficiency of functional dystrophin protein and subsequent loss of myofiber integrity in skeletal and cardiac muscle, thus leaving muscle fibers susceptible to contraction-induced damage [3]. As DMD progresses, muscle turnover and repair cannot keep pace with the constitutive cellular damage that arises, resulting in cell death and replacement with fibrotic and/or fatty tissue (reviewed in [4]).

Phenotypically, dystrophin deficiency is characterized by progressive locomotor-skeletal, respiratory, and cardiac muscle weakness, ultimately resulting in a loss of ambulation as well as respiratory and cardiac failure [5,6,7,8]. Pathologic cardiac manifestations occur in nearly every DMD patient, with the development of cardiomyopathy inevitable by adulthood [9,10] and ultimately progressing to heart failure and mortality in 44–57% of patients [11]. Due to early pharmacological intervention and advancements in respiratory care for DMD patients, cardiomyopathy is now the leading cause of death in this patient population [12,13]. While timely detection and management are essential to slow the progression of dystrophin deficiency-induced cardiomyopathy, uncovering appropriate diagnostic and treatment modalities still remains a challenge due to the early cardiac abnormalities that are often difficult to detect and limitations in cardiac imaging techniques. Additionally, functional measurements in dystrophin deficiency-induced cardiomyopathy are confounded by skeletal muscle weakness, given that typical heart failure symptoms, such as exercise limitations and dyspnea, are masked in this population (reviewed in [5]).

No cure exists for DMD—therefore, treatment is aimed at delaying disorder progression and alleviating symptoms. A comprehensive appreciation of the pathophysiological mechanisms that underlie cardiomyopathy is of the utmost importance to develop targeted treatments. It is worth noting that a single mechanism cannot explain the multifactorial pathogenesis of dystrophin deficiency-induced cardiomyopathy; rather, it is a result of several interrelated mechanisms. While prevailing hypotheses of underlying mechanisms include sarcolemmal weakening, upregulation of pro-inflammatory cytokines, metabolic disruptions, and perturbed calcium homeostasis, mitochondrial dysfunction has been viewed as a potential key contributor [8,14]. Several experimental compounds targeted at the skeletal muscle pathology of dystrophin deficiency are in development, which will ultimately determine whether changes in mitochondrial functions are contributing to myopathy. For now, the effects of these agents on cardiac function remain unclear.

The purpose of this review is to provide an overview of the clinical, physiological, and cellular manifestations of dystrophin mutations in DMD and to describe the theoretical relationships linking these disturbances to mitochondrial dysfunction. The review will conclude with an overview of prospective mitochondrial targets for therapy development.

### 1.2. Epidemiology of Duchenne Muscular Dystrophy

DMD is the most common inherited muscular dystrophy diagnosed in childhood, with an incidence of approximately 1 in 5000 newborn males [15], although variations have been reported elsewhere [16,17,18]. Since DMD is X-linked, it primarily affects males. Approximately 25% and 59% of DMD patients develop cardiomyopathy by 6 and 10 years of age, respectively, with electrocardiogram (ECG) abnormalities and sinus tachycardia (ST) representing the primary early clinical manifestations [5,9,19,20]. By 18 years of age, cardiomyopathy occurs in 98% of patients and progresses to heart failure in 40% of patients [1,21,22]. A higher mortality is linked to dystrophin deficiency-induced cardiomyopathy than to other cardiomyopathies [5].

The prevalence of DMD in females (two copies of the recessive mutation) is extremely rare. Interestingly, one study conducted on aged 22-month-old female C57BL/10 *mdx* mice (dystrophin mutations on both X chromosomes) demonstrated impairments to LV hemodynamic function that was more severe compared to age-matched males and WTs in the absence of differences to cardiac fibrosis, ECG abnormalities, and lifespan between sexes [23,24]. In these mice, it was observed that female *mdx* exhibited heavier heart weight compared to male *mdx* when normalized to body weight [23]. Despite these findings, male *mdx* mice of the DBA/2J background appeared to have heavier hearts compared to age-matched female *mdx* at 8 months old, meaning that the genetic background, sex, and age likely all contribute to the disparate cardiac phenotype that has been observed in DMD mice [25]. The clinical relevance of these findings is uncertain, given that the prevalence of DMD in females is extremely rare. However, female carriers can be either asymptomatic or demonstrate mild to severe skeletal muscle weakness and cardiomyopathy that, like affected males, can worsen with age [26,27].

### 1.3. Genetics

The *DMD* gene is the largest in the human genome [28,29], spanning 2.4 Mb in the Xp21 region [30]. It consists of 79 exons and has several internal promoters that produce an array of dystrophin isoforms expressed in striated and smooth muscles, the brain, the retina, and the kidney [30]. Thousands of mutations in the *DMD* gene have been found to cause dystrophin deficiency [18].

Deletions account for over 70% of mutations and typically result in an altered reading frame that produces a premature stop codon [31,32]. Approximately 20% of dystrophin mutations are point mutations, small deletions, or insertions, whereas duplications make up 5–15% of mutations [33,34]. Deletions and duplications tend to cluster in hotspot regions, located at exons 45–55 and 3–9 on the *DMD* gene [35]. De novo mutations are responsible for one-third of DMD cases [31]. Mutations that disrupt the reading frame or introduce a premature stop codon cause the absence of dystrophin. Mutations that maintain the reading frame allow for truncated dystrophin to be produced, resulting in the phenotype of Becker muscular dystrophy (BMD) [36]. Mutations can potentially confer cardioprotective properties against dilated cardiomyopathy (DCM) (exons 51–52) [37], perpetuate cardiomyopathy (exons 12, 14–17, 31–42, 45, 48–49, and 79) (reviewed in [38]), or be neutral. As mutations can affect different exons on the *DMD* gene, it is unlikely that a single genetic therapy will benefit patients across the spectrum of mutations. Instead, there is substantial value in targeting the numerous secondary cellular and physiological dysfunctions that arise in this disease, including mitochondrial dysfunction.

### 1.4. Dystrophin, Membrane Instability, and Ion Homeostasis

Dystrophin is a rod-shaped protein located beneath the muscle fiber membrane that acts as a molecular shock absorber by transmitting forces generated by sarcomere contraction to the extracellular matrix (ECM) [39]. Dystrophin consists of four major functional domains: an amino-terminal actin-binding domain, a large central rod domain, a carboxyl-terminal domain, and a cysteine-rich domain that binds dystroglycan (reviewed in [40]). Dystrophin is an integral part of the dystrophin-glycoprotein complex (DGC) that connects the intracellular cytoskeleton to the ECM [41] and provides structural support to the sarcolemma. Aside from structural support, dystrophin also serves as a scaffold for proteins involved in various signaling cascades, including sarcolemmal ion channels, to be discussed in detail later [42]. In cardiomyocytes, the Dp427 isoform of dystrophin is expressed and located at the cardiac sarcolemma as well as in the cardiac T-tubules [43]. DMD-induced cardiomyopathy appears to be predominantly attributed to the loss of the Dp427 isoform [43].

The absence of dystrophin results in sarcolemmal membrane fragility and susceptibility to contraction-induced damage, leading to micro-tears in the membrane [40]. In one study examining dystrophin deficiency-induced membrane tears, it was determined through an Evans blue dye assay that left ventricular (LV) cardiomyocyte leakage was significantly increased in C57BL/10 *mdx* mice compared to age-matched control mice [44]. Sarcolemmal membrane fragility, which can also influence ion homeostasis by altering the regulation of sarcolemmal ion channels, is hypothesized to precipitate secondary pathophysiological mechanisms, such as excessive extracellular calcium influx and altered calcium homeostasis, which lead to myocyte degradation and necrosis as well as calcium overload-induced mitochondrial dysfunction [45,46], which will be discussed in detail later. Furthermore, elevated serum cardiac troponin I (cTnI) (as a result of destruction to the myocardium) reflects the increased cellular permeability of the dystrophin-deficient heart [47]. The use of membrane sealants, such as Poloxamer 188, has demonstrated stabilization of the sarcolemma and improvement in acute cardiac function in murine and canine dystrophin-deficiency models [48,49]. Despite the beneficial effects of Poloxamer 188 on acute cardiac function in various mammalian DMD models, several recent studies have identified the deleterious effects of this sealant on dystrophic skeletal muscle, which brings into question its efficacy as a therapeutic intervention [50,51]. This underscores the need to understand how dystrophin mutations cause cardiomyopathy beyond membrane instability alone.

In skeletal muscle, dystrophin mutations also lead to disorganized microtubule organization and oxidized actin, which contributes to weakness and is thought to contribute to mitochondrial dysfunction and other abnormalities [52,53,54,55,56,57], but the degree to which this occurs in the heart remains unknown.

### 1.5. Clinical Manifestations of DMD

The earliest symptoms in DMD patients typically present between 1 and 5 years of age and include a waddling gait, Gower’s maneuver, difficulty climbing stairs, and frequent falls [16,58,59]. Calf hypertrophy tends to develop during early childhood due to the localized accumulation of fatty and fibrotic tissue [60]. Motor skills deteriorate at 6 to 8 years of age when lordosis and scoliosis advance [61]. Scoliosis increases the risk of respiratory failure [62]. Most DMD patients are wheelchair-bound at approximately 13 years of age [63]. Cognitive impairments are observed in roughly one-third of all DMD patients [64]. Secondary to progressive muscle degradation and the leakage of myofiber contents, elevated creatine kinase (CK) levels are detected early, with levels being about 10 times higher in newborns with DMD and approximately 50 to 100 times higher in children with DMD as compared to healthy age-matched boys [27,65].

During disorder progression, cardiac symptoms advance from diastolic dysfunction to myocardial remodeling and fibrosis, leading to contractile weakness and subsequent DCM with decreased systolic function [19,20]. Heart failure and arrhythmias develop gradually, increasing the risk of sudden cardiac death [8]. Largely attributable to advancements in respiratory interventions, such as assistive breathing devices (ventilators), the predominant cause of death has shifted from respiratory failure towards cardiomyopathy (reviewed in [13]), and resultantly, patients may survive well into their third and fourth decades [12,66].

### 1.6. Preclinical Models of DMD

Animal models have been critical in delineating the pathophysiologic mechanisms of dystrophin deficiency and for subsequently trialing therapeutic candidates. The C57BL/10 *mdx* mouse, a genetic and biochemical homolog of human DMD, is the most widely used animal model of dystrophin deficiency and has been invaluable in providing knowledge for therapeutic strategies [67]. The C57BL/10 *mdx* mouse lacks full-length dystrophin due to a nonsense point mutation in exon 23 of the *DMD* gene [67]. This model only exhibits a mild cardiomyopathy late in life that never fully advances into heart failure [68]. Additionally, cardiac fibrosis and dysfunction present very late, if at all, in this model [69]. Although the milder skeletal muscle phenotype and subtle nature of cardiac involvement limit translation to human disorder progression (reviewed in [5]), some features, such as cardiomyopathy, can be unmasked in C57BL/10 *mdx* mice via cardiac stressors, such as ischemia–reperfusion (IR) insults [70,71]. Due to the ease of breeding and maintaining, with a long life expectancy, the C57BL/10 *mdx* mouse models remain a common tool in dystrophin deficiency research [72]. Other mouse models exist with a variety of backgrounds and mutations, but the majority result from the crossing of the C57BL/10 *mdx* mouse with different mutations to worsen the dystrophin deficiency phenotype [73,74]. However, the natural history of cardiomyopathy in each model remains largely uncharacterized.

The DBA/2J-*mdx* (D2.*mdx*) and utrophin–dystrophin double-knockout mice have drawn attention as representing better models of DMD-induced cardiomyopathy, including earlier-onset cardiac deficits [67]. In the D2.*mdx* mouse, cardiac fibrosis is noted at 18 weeks of age [75], but the degree to which cardiac dysfunction occurs at later ages is not understood. Several mouse models for dystrophin deficiency have been developed, but none of the models completely recapitulates the phenotype and disorder progression seen in humans. However, more severe models have a reduced lifespan, are more difficult to breed, and carry an additional mutation not affected in humans, which causes difficulty in reliably extrapolating data to the human phenotype [67,72].

To bridge the gap between small rodent (mouse) DMD models and the human phenotype, several different rat DMD models have been investigated. To date, the majority of rat DMD models have been developed to genetically target the genomic region that spans exons 3–26 [76,77,78]. This is a major limitation in the model, given that a major mutation hotspot for the classical, progressive DMD phenotype is between exons 45 and 55 [76]. For this reason, a new rat DMD model was generated with a deletion mutation in exon 52 on the rat *DMD* gene, known as the R-DMDdel52 rat [76]. In comparison to the *mdx* mouse model, the R-DMDdel52 rat exhibits remodeling of the entire striated musculature, including both the heart and diaphragm, which culminates leading to premature lethality between 10 and 14 months of age [76].

The golden retriever muscular dystrophy (GRMD) model offers remarkable resemblance to the progressive locomotor, respiratory, and cardiac muscular phenotype observed in human DMD patients [79]. Similar to humans, these canines are characterized by highly variable cardiomyopathy progression, making the assessment of therapies challenging [80,81]. Unfortunately, high costs, limited supply, and difficulty in maintaining a colony preclude the widespread use of GRMD models [72].

In order to more closely recapitulate the human DMD phenotype (in comparison to small rodent models) without limitations imposed by the financial burden of large mammals (e.g., GRMD), a DMD rabbit model was developed by cytoplasmic microinjections of Cas9 mRNA and single-guide RNA (sgRNA) [82]. The rabbit DMD model was developed by targeting exon 51 (commonly mutated in human DMD patients) to disrupt the open reading frame of *DMD* in rabbits, thereby generating *DMD* KO (knock-out) rabbits [82]. This model exhibits many hallmark pathologies of DMD, including elevated serum creatine kinase, impaired mobility/ambulation, muscle necrosis and regeneration, and, importantly, cardiomyopathic manifestations (increased cardiac fibrosis, impaired function, etc.) [82]. Although further research is still required to comprehensively validate and characterize this model, the clear evidence of cardiomyopathy at 4 months of age (demonstrated by reduced left ventricular ejection fraction and fractioning shortening), paired with the close resemblance to its human DMD counterpart, makes this an attractive model for preclinical drug screening studies [82].

A dystrophin-deficient pig model that is characterized by a targeted deletion of *DMD* exon 52 has also been recently developed to recapitulate human size, anatomy, and physiology more closely [83]. Of relevance, pigs with deletions to *DMD* exon 52 do not express dystrophin in skeletal muscle and have elevated serum creatine kinase, impaired function including mobility, muscle weakness, and a maximum life expectancy of 3 months due to severe deteriorative respiratory impairments [83]. This model is also noted for its cardiac dystrophin deficiency, with several notable downregulations to components of the dystrophin-associated protein complex that have been observed [84].

Aside from mammalian models that provide integrative insight into the pathophysiology of DMD, work from the nematode worm *Caenorhabditis elegans* can also provide invaluable information regarding disease progression. This double-mutant model contains a combination of a dystrophin homologue (*dys-1*) null mutation and a weak mutation to *hlh-1*, which encodes for a MyoD homologue [85,86]. These worms are interesting given that they exhibit a time-dependent loss of motility as well as muscle degeneration, which are similar observations to those seen in *mdx* mouse mutants [85]. The worms are typically used to identify new components of the dystrophin–glycoprotein complex and also unidentified suppressors of muscle degeneration [85].

Induced pluripotent stem cells (iPSCs), another preclinical model for dystrophin deficiency, are reprogrammed from patient-specific somatic cells and carry the same genetic defects as DMD patients [87]. These cells make it possible to obtain functional cardiomyocytes from DMD patients and offer an important complement to animal models in studying dystrophin deficiency (reviewed in [88]). However, iPSCs do not manifest a fully mature phenotype, which may limit their utility in pre-clinical research [89].

Collectively, most of the research demonstrating mitochondrial dysfunctions in response to dystrophin mutations has been conducted in the C57BL/10 *mdx* and D2.*mdx* mouse, as well as some literature in males with DMD. These findings are discussed in Section 3.

### 1.7. Current Standard of Care and the Need for New Therapies

No cure exists for DMD or its associated cardiac dysfunction, and therefore, most interventions are aimed at treating the symptoms and delaying the progression of dystrophic cardiomyopathy. Unfortunately, cardiac transplantation is generally contraindicated in DMD patients due to muscular weakness and respiratory insufficiency [90]. Left ventricular assist devices (LVAD), if implemented as a destination therapy rather than a bridge to transplantation, can be considered as a therapeutic option in DMD patients [91]. While novel therapies have been in development to specifically address defects resulting from dystrophin deficiency, their effectiveness with respect to cardiac function has been limited due to the diverse range of genetic mutations in DMD and the complex risks of each prospective therapy (reviewed in [92]). For example, certain small molecule-based therapies used in practice or under development target secondary cellular dysfunctions that are common in most, if not all, DMD patients and therefore have the potential for widespread use. Such therapies tend to focus on symptom management and delaying the development of cardiomyopathy and heart failure. Other gene-based treatments that focus on restoring dystrophin expression in a truncated or complete sequence must be customized to each unique mutation that occurs in DMD.

The most widely used intervention in DMD is corticosteroids, which have been proven to reduce inflammation, prolong strength, and delay the loss of ambulation by 1 to 3 years in DMD patients [93]. Therapy most often involves up to daily dosing of either prednisone or deflazacort initiated around 2 to 5 years of age [32]. Although it is unknown how steroids delay disease progression, likely mechanisms include reduced cytokine production and the activation of insulin-like growth factors, decreased myocardial inflammation and fibrosis, cell membrane stabilization, increased myoblast proliferation, improvements to skeletal muscle function, and upregulation of proteins that support dystrophin [94,95]. Studies in DMD patients have collectively revealed the protective benefits of steroid treatment in dystrophic hearts, including decreased fibrosis, preserved ventricular function, and better survival [95,96,97,98,99,100].

When steroid therapy is initiated prior to the onset of cardiomyopathy, delayed development of cardiac dysfunction has been observed [101]. Data from a surveillance program demonstrated that a longer duration of steroid use correlates with a greater improvement in LV function [95]. Furthermore, the duration of steroid treatment has been proven to be inversely correlated with the incidence of cardiomyopathy in DMD patients [95]. Long-term steroid treatment beyond the loss of ambulation in DMD led to improvement in all-cause mortality secondary to improved cardiac outcomes [96]. On a molecular level, a study in 7-week-old C57BL/10 *mdx* mice demonstrated that glucocorticoids prevented calcium-induced mPTP opening and restored ADP-stimulated respiration as the result of enhanced expression of Complex III, Complex IV, and IV protein content markers in skeletal muscle [102], but further work on cardiac muscle is required in this area.

More recent studies have suggested that pulsed steroid regimens may limit the adverse effects while maintaining the benefits of daily steroid dosing in DMD patients [103,104]. Preclinical studies have demonstrated that steroids may worsen the progression of cardiomyopathy in the heart of C57BL/10 *mdx* mice, with findings including decreased cardiac function, increased dilation, and increased cardiac fibrosis [105,106,107], consistent with similar findings in the diaphragm [68]. It should be noted that these studies used a more continuous method of drug delivery than is equivalent to the single daily dose therapies administered in DMD patients [99].

Chronic steroid use may activate mineralocorticoid receptors (MR), which likely cause adverse effects, including reduced bone density, obesity, hypertension, adrenal insufficiency, and increased muscle catabolism [108,109,110]. MR antagonists, such as Eplerenone and Spironolactone, are treatments for heart failure with reduced LVEF and are used in some cases of DMD-induced cardiomyopathy [30].

Additional therapeutic options such as angiotensin inhibition, beta-adrenergic receptor blockers, and Tamoxifen—which is a first-generation selective estrogen receptor modulator (SERM)—have demonstrated some potential for DMD but are not approved for clinical use, while gene-targeted therapies are limited. Angiotensin II and angiotensin II type 1 receptors induce many harmful cardiac effects, including increased fibrosis, remodeling, ROS production, and cardiomyocyte death [111,112,113]. The two angiotensin-inhibiting drug classes used in heart failure and DMD patients are angiotensin receptor blockers (ARBs) and angiotensin-converting enzyme inhibitors (ACEIs). ACEIs prevent an enzyme from converting angiotensin I to angiotensin II, which decreases the activation of genes that enhance fibrosis and scarring of the myocardium [114] and block the action of angiotensin II, allowing veins and arteries to dilate. The activation of beta-adrenergic receptors enhances heart rate elevations and increases contractility through its action on the calcium transients in the cardiomyocyte [115]. Beta-blockers may limit these adverse effects through inhibition of beta-receptor binding and subsequent catecholamine binding [115]. Beta-blockers are considered a second-line candidate for cardiac-related therapy in DMD patients, typically administered in addition to an angiotensin-inhibiting agent [5,116].

## 2. Cardiomyopathy in DMD: Functional and Histological Manifestations

As previously noted, typical cardiac symptoms of exercise intolerance and dyspnea are masked in DMD patients due to the influence of skeletal muscle weakness on these same parameters [5]. Nonetheless, the detection of early changes in the structure and function of the heart is crucial to initiating timely treatment in hopes of yielding better outcomes. Current guidelines recommend yearly cardiac screening at diagnosis [27].

### 2.1. Echocardiography

Echocardiography is recommended until at least 6 to 7 years of age, when the child can lie still without anesthesia, at which point cardiac magnetic resonance imaging (CMR) is recommended due to its ability to detect subtle cardiac changes prior to overt cardiac dysfunction [27]. Transthoracic echocardiography (TTE) is a non-invasive, readily available diagnostic tool, but measurement of the left ventricular ejection fraction (LVEF) using standard TTE rarely detects abnormal cardiac function in DMD patients within the first decade of life [117,118]. Two-dimensional fractional shortening (FS) and 5/6 area-length LVEF were found to be the most accurate and reproducible objective measures of left ventricular (LV) function using TTE in DMD patients [117]. However, TTE has been observed to underestimate LV function compared to CMR [117,119]. Myocardial performance index (MPI) and Doppler tissue imaging (DTI) may be capable of detecting myocardial dysfunction prior to the development of systolic dysfunction [99]. MPI, an assessment of global heart function, has been observed to correlate with EF [120]. However, in DMD patients, calculation of the MPI showed abnormalities in 79% of patients, whereas 40% of patients had an abnormal LVEF on TTE [121]. DTI, which does not require good resolution, can detect early changes in the development of cardiomyopathy through measures of myocardial tissue velocities and strain [99]. Decreased tissue velocities have been observed in asymptomatic patients as young as 8.8 years and were able to predict poor outcomes with 85% accuracy [122]. Reduction in peak systolic radial strain has been observed in the posterior wall of DMD patients [123], commonly seen in the outer portion of the wall [124] and in those with normal systolic function [125]. DTI-derived strain measurement is less reliable in DMD patients due to their skeletal deformities obscuring proper Doppler beam placement. Thus, speckle tracking echocardiography (STE) is preferred to measure 2D strain. STE can detect subclinical LV dysfunction prior to a decrease in LVEF through measurement of myocardial strain [126,127] and is able to assess segmental and global myocardial function in the longitudinal, radial, and circumferential displacements [128]. Before the appearance of overt cardiomyopathy in patients with DMD, a significant reduction in the global LV STE strain has been reported [127]. Myocardial strain, as measured by STE, was observed to be abnormal in approximately 50% of DMD patients, even in those with normal EF, suggesting that strain may identify early cardiac involvement [129,130]. Lower global longitudinal strain (GLS) values were seen in DMD patients, with a decrease of 0.34% per year according to age [127]. The magnitude of the difference in strain was greatest in the inferolateral and anterolateral segments [127,131]. Studies have reported variable findings using STE, as the longitudinal peak systolic strain was reported to be more pronounced in the apical area and mid-anterior segment in one study [132], while another study revealed more prominent changes in the basal lateral segments [131] with decreased peak systolic strain [133]. In addition, a non-controlled study found that circumferential STE strain correlated moderately well with CMR strain and reported a trend toward reduced STE strain in patients with late gadolinium enhancement, a sign of fibrosis on CMR [127]. This could represent a good cardiac assessment option in DMD children too young to undergo CMR without anesthesia.

### 2.2. Limitations of Echocardiography

Patients with DMD have poor acoustic windows due to increased adiposity, altered body habitus, lung hyperinflation, and limited mobility [134,135]. These factors degrade echocardiographic image quality and negatively impact interpretation [119]. As a result, in several studies, echocardiography has been deemed inadequate for detecting cardiac involvement in DMD patients in the first decade of life and sometimes beyond [119,136]. Additionally, TTE cannot perform tissue characterization to detect early myocardial fibrosis [137] and is unable to accurately account for the regional distribution typically apparent in DMD-induced cardiomyopathy. It was observed that TTE misclassified 20% of DMD patients, and 37% of the myocardial segments were unable to be visualized compared to CMR [117,119]. Despite these limitations, more advanced imaging may be contraindicated, and TTE still remains valuable in the assessment and monitoring of DMD patients.

### 2.3. Cardiac Magnetic Resonance Imaging

CMR is a non-invasive imaging tool that provides accurate volumetric measurements, assessment of wall motion abnormalities, and tissue characterization without negative effects from body habitus or other similar external factors [117]. Advantages to this technique include its excellent reproducibility and operator independency [138]. CMR has been proven efficient in detecting myocardial fibrosis in patients with DMD [116,125]. CMR in DMD patients has demonstrated a better diagnostic yield for detecting preclinical features of cardiomyopathy and greater sensitivity for identifying subtle changes in cardiac function, wall motion, and structural abnormalities than TTE [137]. LVEF, as derived from CMR, was found to correlate poorly with LVEF derived from TTE, and CMR-derived LVEF was found to moderately correlate to TTE-derived FS [117], even with adequate imaging quality [139]. As such, CMR is now the preferred cardiac imaging technique for patients with DMD [27]. Even in the absence of overt cardiomyopathy, CMR has identified a pattern of fibrosis in female carriers similar to that observed in DMD patients [138]. The detection of LV wall motion abnormalities on CMR has demonstrated good predictive value for the presence of regional cardiac dysfunction [140], whereas the transmural pattern often located at the inferolateral wall independently predicts adverse cardiac events in DMD patients with normal LVEF [141].

Late gadolinium enhancement (LGE), a sensitive marker of myocardial fat and fibrosis, appears as a result of diminished contrast washout on CMR [137]. This tool can be used to detect subclinical cardiac disease prior to LV dysfunction and may predict adverse cardiac events in DMD patients [136,141,142]. In DMD-induced cardiomyopathy, LGE is subepicardial and absent in the subendocardium [143]. LGE appears to correlate with age such that patients with LGE tend to be significantly older with decreased LVEF [136,144]. In fact, the presence of LGE correlated with a 2.2% decline in LVEF per year [145]. LGE in DMD patients has also been linked to greater LV dilation and dysfunction, as well as a higher incidence of arrhythmias [146]. LGE was reported in 30% of patients with normal LVEF and in 84% of patients with abnormal LVEF [136]. One study revealed that patients with normal function on TTE had LGE on CMR and that several segments with abnormal wall motion by CMR were not detected on TTE [117]. LGE has been observed in patients under 10 years [99,119], predominantly in the inferoseptal and anterolateral segments, but it also typically involves the basal inferolateral free wall [137]. In addition, LGE of the subepicardium of the lateral LV wall is commonly observed, with intramural septal LGE becoming more prevalent with disorder progression [141,147]. It appears that the presence of LGE can predict the severity of cardiomyopathy in DMD [145] and correlates to a higher risk of arrhythmias in DMD [146]. LGE may guide therapeutic interventions, as the most recent DMD treatment guidelines recommend initiating cardiac therapy when LGE presence is first observed [143].

Strain imaging quantifies regional tissue deformation, and its measurements are potential early markers of cardiac involvement in DMD [148]. The CMR strain correlates more closely with CMR LVEF compared to the TTE strain [119]. A sensitive marker used to identify cardiac pathology is the peak circumferential strain (ε*cc*) at the mid-ventricular level, which can also assess the effects of therapeutic interventions on cardiac function and measure the degree of cardiac pathology [100,131,136,144,149,150,151]. This measure was able to detect cardiac pathology as early as 5 years of age, and the findings of this study revealed the heterogeneity in cardiac progression in patients with DMD [151]. Additionally, the circumferential uniformity ratio estimate (CURE) was identified in this study to have potential as a tool for understanding cardiac pathology in the DMD population [151]. Global ε*cc* appears to be the most sensitive strain marker for subclinical myocardial changes in DMD [149,152], with more prominent differences in anterolateral, inferolateral, and inferior segments [153]. Reduced myocardial strain has been observed in the presence of normal LVEF [144,150,154]. Specifically, decreased global circumferential and segmental strain, as well as mitral annular plane systolic excursion (MAPSE), has been observed in DMD patients with normal LV function and absence of LGE [150,152]. Similarly, decreased LV myocardial peak circumferential strain was observed in DMD patients under 10 years of age and declined with age [144]. In DMD patients who develop overt LV dysfunction, CMR strain was found to be significantly worse than in those who do not develop cardiomyopathy [149].

In cases where contrast is contraindicated, such as patients with vascular access issues, renal insufficiency, or contrast allergies, non-contrast CMR techniques are important [119]. CMR-feature tracking does not use contrast and has been able to differentiate DMD patients from controls [153,155]. Without contrast, ε*cc* can still detect myocardial global and regional alterations [144,150,156].

Other CMR tools are T1 and T2 mapping. T1 mapping, which measures diffuse myocardial fibrosis and extracellular volume expansion, can identify early myocardial fibrosis even in the absence of LGE [157]. Both pre- and post-contrast T1 mapping can detect earlier and more subtle signs of cardiac dysfunction [158]. In contrast to LGE, which demonstrates focal fibrosis, T1 mapping can identify diffuse fibrosis in cardiomyopathy [159]. T1 values are increased in DMD patients and thus may serve as early markers of myocardial fibrosis [160]. T1 mapping is often unable to differentiate fibrosis from inflammation or fat infiltration, which may limit its use in DMD patients [161,162].

T2 mapping may be able to identify fat infiltration, edema, and inflammation within the affected muscle [158]. In a study using T2 mapping in DMD boys, the full-width half-max, a measure of T2 heterogeneity, correlated well with reduced LVEF and circumferential strain [163].

### 2.4. Limitations of Cardiac Magnetic Resonance Imaging

The disadvantages of CMR include expense, limited availability, patient discomfort due to position and immobility, claustrophobia, length of study, and the need for sedation in some patients [117]. The presence of hardware, such as spinal rods to treat scoliosis, may distort the image, limiting the use of CMR in DMD patients [164]. With repeated exposure, gadolinium has been reported to accumulate in organs and may lead to toxicity [165,166,167]. In addition, the breath-holding segment of this procedure has been a limitation for the DMD population, but advancements have decreased breath-holding times, and free-breathing sequences are sometimes an option [143,168]. Despite these limitations, CMR is a powerful imaging modality for monitoring cardiac involvement in DMD patients, and circumferential strain and LGE are sensitive tools that can be utilized to detect occult cardiomyopathy prior to LVEF decline [143].

### 2.5. Hemodynamic Biomarkers

Biomarkers indicative of DMD are suspected to be released into circulation as a result of sarcolemmal tearing, a phenomenon that that occurs in response to the mechanical stress of contraction caused by compromised membrane integrity arising from the dystrophin mutation. Cardiac troponin I (cTnI) and T (cTnT) are proteins that comprise the contractile unit of myocardial cells and are biomarkers for damage to the myocardium [169]. Serum cTnT can be elevated in the setting of neuromuscular disease and has been shown to correlate better with creatine kinase and myoglobin levels than with cTnI [170,171]. Therefore, cTnI is more specific for myocardial injury in DMD as it is not expressed in skeletal muscle during regenerative processes [172,173]. Cases of acutely elevated cTnI in DMD patients have been reported with abnormal ECGs but no evidence of coronary artery disease [174,175,176,177,178,179]. Heterogeneity of cTnI levels has also been reported throughout the literature [169]. Significantly elevated levels of cTnI have been reported in DMD patients with mild LGE compared to those without LGE. However, cTnI levels were not elevated in patients with moderate-to-severe LGE. Since LGE is representative of cardiac fibrosis on CMR, this finding is likely due to increased cell death and fibrofatty tissue replacement in early disease and decreased enzyme leak at later stages of the disease when fibrofatty tissue has replaced most of the myocardium [180]. Elevated troponin levels have also been reported in asymptomatic patients prior to the development of cardiac disease [180], which may represent the necrosis and fibrosis resulting from subclinical or early cardiac remodeling. Decreased cTnI was observed in dystrophic dogs treated with membrane sealant, which was thought to ameliorate cardiac injury [49]. No current recommendations exist for routine monitoring of cTnI in DMD patients. Therefore, only symptomatic patients tend to undergo testing, making unbiased assessment of the distribution of troponin levels difficult [169].

In addition to troponin, B-type natriuretic peptide (BNP) is a well-known serological marker of cardiac dysfunction; however, studies have revealed that BNP levels are inconsistent and thus are not appropriate measurements of cardiomyopathy in DMD patients [181]. Despite this, lower BNP levels have been reported in patients with DMD-induced cardiomyopathy versus those with idiopathic cardiomyopathy [182]. Only a mild elevation of BNP levels is seen, typically only in the presence of severe cardiac dysfunction [183,184].

Other potential biomarkers and indicators include plasma alpha-ANP, CK-MB, and diastolic abnormalities. Elevation of plasma alpha-ANP levels may be an indication of poor prognosis in DMD patients as it is correlated with congestive heart failure and respiratory failure [185]. Since CK-MB, which is present in both skeletal and cardiac muscle types, can be seen during skeletal muscle regeneration, this protein is not a good marker for cardiac involvement in DMD patients as it can also be indicative of damage to other muscles in the body [186]. Furthermore, DCM and systolic dysfunction are preceded by diastolic dysfunction in DMD, and therefore, diastolic abnormalities may serve as early indicators for early cardiac decompensation [97].

### 2.6. Systolic and Diastolic Dysfunction

Several studies have reported the presence of systolic and diastolic dysfunction in patients with DMD. It is hypothesized that the calcium dysregulation resulting from the absence of dystrophin presents phenotypically as sustained ion-driven myocyte contraction, which appears as underfilling of the LV and impaired relaxation prior to the development of DCM [97,187]. The authors of [187] described the heart in this phase as “tonic contraction”, a term describing the abnormally increased tone without the ability to relax. Although the LV appears smaller due to its inability to relax, the mass is unaltered. These changes likely lead to a decrease in stroke volume, which may be correlated with the resting tachycardia often present in DMD patients [187]. Furthermore, impaired relaxation (which may be attributed to increased myocardial fibrosis, leading to stiffness of the ventricular walls), diminished the LV cavity size at the end of diastole (a possible outcome of impaired calcium handling to be discussed later), and thickened myocardium have been observed in DMD animal models [187,188,189,190]. Prior to the appearance of clinical symptoms and decreased LVEF in DMD patients, early changes in diastolic function were observed through DTI [133]. Specifically, abnormal measures of impaired diastolic function included altered myocardial velocity E’ basal lateral and the mitral valve velocity/myocardial basal lateral E-wave ratio E/E’ [191,192]. In a recent study utilizing CMR in DMD patients, LV atrioventricular plane displacement (LVAPD), a marker of systolic and diastolic dysfunction, was decreased in patients with DMD, indicating reduced atrioventricular (AV) plane displacement [151]. Decreased AV displacement correlates to a reduction in contractility and relaxation of the heart, resulting in reduced diastolic filling pressure and diastolic volume [151]. This study found that LV end-diastolic volume (LVEDV) was significantly lower than controls, an observation supported well by the literature [127,152,193]. Other significant findings in the study by [151] included a reduced LV end-diastolic index (LVEDI) and changes in LV mass and LV end-systolic volume (LVESV).

### 2.7. Cardiac Fibrosis

The deterioration of cardiomyocytes observed during dystrophin deficiency-induced cardiomyopathy activates an inflammatory cascade that results in macrophages clearing damaged cells and fibroblasts invading the compromised area to form fibrotic (collagenous) scar tissue. In DMD, a prolonged subclinical phase of myocardial fibrosis is typically observed, beginning early in life [97].

The death of cardiomyocytes and subsequent development of cardiac fibrosis initially present in the posterobasal segment of the LV [194,195] and extend to the outer third and ultimately the entire LV and septum [196]. The fibrotic lesions are more prominent in the subepicardium and spare the muscle fibers nearest to the chambers in GRMD [197]. Additionally, fibrosis of the papillary muscles and posterobasal area contributes to mitral valve regurgitation in DMD [198]. Fibrosis is prevalent from a young age, as evidenced by LGE on CMR [99,191], where cardiac fibrosis was identified in 17% of patients under 10 years, 34% of patients between 10 and 15 years, and 59% of patients over 15 years [136]. Fibrosis occurs prior to the onset of decreased systolic function, which has been demonstrated with LGE on CMR [191]. Myocardial fibrotic changes typically have a heterogeneous distribution, according to evidence of regional wall motion abnormalities determined via imaging modalities [119]. While widespread scarring leads to stretching and thinning of the myocardial walls, focal fibrosis increases the risk of sudden death [158].

### 2.8. Dilated Cardiomyopathy

As myocardial fibrosis increases with disorder progression, the fibrotic region of the heart gradually stretches and thins, losing its contractility and resulting in DCM (reviewed in [40]). This may involve dilation of the LV or both ventricles, although right ventricle (RV) function tends to be relatively preserved in the setting of improved respiratory interventions [199]. Dystrophin deficiency-induced cardiomyopathy is different than other types of cardiomyopathy in that the LV dilatation is less prominent, while the prognosis is worse [90]. DCM typically occurs in the second decade of life in DMD patients, although it has been reported in children under six years of age [9]. The clinical signs of DCM in DMD patients include increased LV diameter and volume, reduced FS, decreased LVEF, and development of mitral valve regurgitation [40].

### 2.9. Arrhythmia

Frequent ECG changes detected in DMD after the development of cardiac fibrosis consist of ST, short PR interval, inferolateral Q waves, increased R/S ratio in precordial leads with tall R waves, left atrial abnormality, and right axis deviation [164]. In addition to dystrophin deficiency-induced cardiac fibrosis, calcium transients, and elevated ROS, ECG abnormalities may be secondary to dysregulated sodium, calcium, and potassium channels, kinases, and nitric oxide synthase (nNOS) triggered by the absence of dystrophin [200]. Briefly, dystrophin is thought to play an important role in scaffolding voltage-dependent sarcolemmal ion channels in the heart via syntrophin binding, which is an adaptor protein that binds directly to two sites in dystrophin’s carboxyl-terminal region [200]. Dystrophic cardiomyocytes have demonstrated considerable sodium channel loss-of-function, while the activity of some potassium channels may also be reduced [200]. It is hypothesized that sarcolemmal ion channel abnormalities may occur prior to the onset of cardiomyopathy in the dystrophic heart and may represent a primary effect of the *DMD* mutation [200]. QRS duration appears to increase with age, regardless of systolic function [201]. The onset of resting ST is typically seen in DMD patients by 5 years of age, and conduction changes are observed by 10 years of age [40]. Cellular mechanisms, including altered calcium homeostasis and elevated ROS, are hypothesized to cause arrhythmias in these patients and will be discussed in more detail in the following sections. More serious arrhythmias, such as atrial fibrillation, AV block, ventricular tachycardia, and ventricular fibrillation, have been reported in the presence of advanced fibrosis [40,202]. One study estimated that approximately 44% of DMD patients had arrhythmias, including frequent atrial (3%) or ventricular premature contractions (22%), atrial (16%) or ventricular couplets (32%), supraventricular tachycardia (SVT) (9%), and ventricular tachycardia (VT) (13%) [203]. Arrhythmias have been shown to be significantly correlated with decreased systolic function [203,204], and an age older than 17 years was significantly associated with the development of SVT or VT [203]. Sudden death does occur in DMD patients, but the percentage due to arrhythmias is unknown [205]. Continuous Holter recordings have shown resting ST, loss of cardiac circadian rhythm, and heart rate variability [206]. Although one study showed that ECG abnormalities frequently precede cardiac dysfunction by several years [207], no differences were noted in the ECG of DMD patients with cardiomyopathy compared to those in the subclinical stages [208]. These inconsistencies suggest that ECG is not a useful diagnostic tool; however, periodic Holter monitoring is recommended in this population [27].

### 2.10. Cardiac Chamber-Specific Differences in DMD

To date, the majority of research in the hearts of DMD patients and dystrophin-deficient pre-clinical models has been conducted specifically in the LV or, more commonly, in the unspecified “whole heart”. A limited data set now suggests that dystrophic cardiomyopathy may affect the heart heterogeneously, with differing data on histopathology (e.g., fibrosis, calcification, etc.), function (e.g., echocardiography, hemodynamics, etc.), and other related parameters observed across ventricles. Several studies across both human and pre-clinical DMD models have determined differences in ejection fraction, fractional shortening, and other longitudinal measurements between ventricles [191,209,210]. The authors of [75] demonstrated that 18-week-old D2.*mdx* hearts exhibit thick epicardial fibrotic-calcinosis layers that are limited and restricted to the RV, but such changes are not detectable in the LV or septum [211]. While systolic dysfunction is evident in both ventricles of the D2.*mdx* heart, it was determined that RV function was slightly better than LV despite the elevated fibrosis [211]. Together, these data suggest that RV damage potentially precedes the onset of significant cardiac complications [211]. Historically, the role of the RV in dystrophin deficiency-induced cardiomyopathy has been underappreciated in place of the larger, more physiologically demanding LV. For this reason, the literature comparing pathology across ventricles is limited. Although, to date, some work has been conducted to elucidate RV function and its role among the respiratory impairments associated with dystrophin deficiency, further research is required to investigate parameters such as hemodynamics and echocardiography in young dystrophin-deficient rodent models (mice aged >24 weeks). Additionally, there is an abundance of conflicting data with regard to ventricle-specific disease progression due to the use of different dystrophin-deficient models and time points. Future research should seek to track the time-course progression of ventricular abnormalities in an established dystrophin-deficient pre-clinical model to determine the order in which these pathologies arise. Additionally, the field would invariably benefit from building on pre-existing literature that examines how LV mitochondrial perturbations correspond to LV functional differences by extending experiments to the RV for chamber-specific comparisons.

## 3. Inflammation, Calcium Dysregulation, and Their Contribution to Mitochondrial Dysfunctions in DMD

### 3.1. Inflammatory Signaling and Immune Response

Although inflammation is necessary for healing, it can also be damaging if it occurs aberrantly. Inflammatory and immune infiltrate are responsible for clearing damaged muscle cells and are often present prior to the onset of symptoms. The immune infiltrate primarily consists of macrophages and T cells in young (2 to 8 years of age) DMD patients [212,213]. Since the heart has a limited capacity to regenerate, macrophages secrete chemokines, such as transforming growth factor beta (TGF-β), to activate resident fibroblasts and endothelial cells in response to injury, thus increasing the ratio of nascent myofibroblasts to quiescent fibroblasts. The fibroblasts are activated into collagen-secreting myofibroblasts and enhance ECM deposition at the injury site, forming fibrocollagenous scar tissue within the wall of the ventricular myocardium [214,215]. Muscle biopsies from DMD patients demonstrated that TGF-β expression is associated with skeletal muscle fibrosis [216]. A high concentration of macrophages and T lymphocytes has been reported in pooled dystrophic muscles (quadriceps, hamstrings, and gastrocnemius) beginning early in disorder progression, suggesting that these cells play a crucial role in the pathology of dystrophic muscle [213,217]. Sarcolemmal lesions in dystrophic skeletal muscles initiate calcium leakage and inflammatory cytokine upregulation, produce and release tumor necrosis factor alpha (TNF-α) and histamine, and contribute to mast cell degranulation and elevations to eosinophils [213,218,219]. A combination of these cytokines contributes to a proinflammatory environment and consequently promotes muscle necrosis [213]. While inflammatory mechanisms have been discovered in dystrophic skeletal muscle, these pathways have not been extensively examined in the dystrophic heart, thus representing a future avenue of research.

Two main inflammatory pathways are involved in DMD: the nuclear factor kappa-light-chain-enhancer of activated B cells (NF-κB) pathway and the nucleotide-binding oligomerization domain (NOD)-like receptor family pyrin domain containing 3 (NLRP3) pathway. NF-κB, a transcription factor that regulates the expression of chemokines and cytokines, may be activated by dystrophin deficiency-induced mechanical stretch. Increased activation of NF-κB is seen in dystrophic skeletal muscle [213,220]. The overexpression of chemokines is hypothesized to occur prior to initial disorder onset (defined as the initial mechanical damage that results from the absence of the membrane-stabilizing effects of dystrophin) and induce macrophage and T-cell infiltration [213,221]. It has been previously determined that the inhibition of NF-κB in utrophin/dystrophin-deficient mice improves cardiac contractile function [222].

Inflammatory cytokines like IL-6 are elevated in both pre-clinical models [223] and humans [213,224,225] with DMD. The potential relationships between inflammation and mitochondrial dysfunction are notable given that TNF-α and IL-6 are examples of cytokines that induce decreases in mitochondrial oxidative phosphorylation and elevations in mitochondrial superoxide production in a variety of cell types and models unrelated to dystrophin mutations [226,227,228]. Indeed, the current standard of care involves immunosuppression with glucocorticoids [93], which slows the progress of the disease through mechanisms that are thought to be related, in part, to reduced inflammation [104]. Nonetheless, the remaining influence of inflammation is thought to contribute to mitochondrial dysfunctions in muscle [14], although this is not fully characterized in the context of disease and glucocorticoid interactions.

### 3.2. Calcium-Handling Dysregulation and Cell Death

In healthy cardiac muscle, contraction and relaxation are driven by the cycling of calcium between the sarcoplasmic reticulum (SR) and cytoplasm. Briefly, this process involves plasma membrane depolarization that subsequently activates the L-type calcium channels (LTCC), which allows calcium to flow into the cell and subsequently causes a larger calcium flux from the SR through the ryanodine receptor 2 (RyR2), resulting in muscle contraction. This process is termed calcium-induced calcium release (CICR). It is imperative to note that the relationship between muscular dystrophy-induced elevations to net intracellular influx of calcium and muscle-cell necrosis was first proposed 40 years ago [229,230]. It was hypothesized that this increased intracellular calcium concentration triggers a vicious cycle of downstream mitochondrial calcium overload and eventual insufficient ATP synthesis, which further perpetuates cytoplasmic calcium levels by impairing calcium pumps, leading to the hypercontraction of muscle fibers and subsequent cell necrosis [229]. Interestingly, Wrogemann and colleagues were among the first to determine that calcium chelators such as EDTA could ameliorate the depressed respiration in dystrophic mitochondria, at least from skeletal muscle, by lowering mitochondrial calcium concentrations [229]. Although this finding was not originally established in dystrophic cardiac tissue, it is still a seminal finding that highlights the importance of intracellular calcium balance in dystrophic tissue and how calcium handling can be targeted in pursuit of a therapy.

Multiple mechanisms are believed to contribute to calcium overload and disrupted calcium homeostasis. Increased calcium influx and resting levels of mitochondrial calcium have been observed in myocytes from C57BL/10 *mdx* mice [231]. In addition to the extracellular calcium that leaks through the sarcolemmal membrane tears, ion channels and calcium handling proteins likely contribute to the excessive calcium found in dystrophic myocytes. Stretch-activated calcium influx differences, increased diastolic calcium levels; altered calcium transient kinetics; and changes in calcium-handling protein expression, activation, or post-translational modifications have been observed in cardiomyocytes of C57BL/10 *mdx* mice [46,48,232,233] (Figure 1).

Excessive cytoplasmic calcium may activate calcium-dependent proteases, such as calpains, contributing to apoptosis [234]. A study investigating single ventricular myocytes from C57BL/10 *mdx* proposed that calcium-activated calpains degrade troponin I, resulting in contractile dysfunction and decreased calcium sensitivity [232]. A separate study assessing calpains in 4-week-old C57BL/10 *mdx* mice determined that total concentrations of calpains were elevated in necrotic hind limb muscle from dystrophic mice compared to controls; however, further research on the heart is required [235].

Another possible source of calcium overload in dystrophic cardiomyocytes is through transient receptor potential channels (TRPC) and the LTCC Ca_v_1.2 [236,237,238,239]. TRPCs are hypothesized to engage in the augmented stretch-activated cation influx seen in C57BL/10 *mdx* cardiomyocytes, and overexpression of these channels has been observed in dystrophic cardiomyocytes [236,237,238,239]. A decrease in excessive calcium was demonstrated in C57BL/10 *mdx* cardiomyocytes using inhibitors of TRPC, stretch-activated channels, and transient receptor potential vanilloid (TRPV2) [232,238]. Redox modifications of Ca_v_1.2 alpha1 subunit that occur during periods of elevated oxidative stress lead to increased channel-mediated calcium influx [200,240,241]. Resultantly, ROS may partly explain the gain-of-function calcium channel abnormalities observed in dystrophic cardiomyocytes [200]. Indeed, communication between LTCC and mitochondria has proven to be crucial in healthy cells for metabolic function, and this communication was disrupted in C57BL/10 *mdx* cardiomyocytes [231,242]. Studies in mice lacking either dystrophin alone or dystrophin and utrophin demonstrated increased calcium influx through the LTCC in cardiomyocytes [232,238]. The LTCC pathway to calcium overload is also suspected to play a role in the calcium-dependent arrythmias seen in DMD [243].

RyR2 and sarcoplasmic/endoplasmic reticulum calcium ATPase (SERCA2a) may enhance the release of store-operated calcium in cardiomyocytes of DMD patients [72,244,245,246]. It has been observed that in C57BL/10 *mdx* hearts, RyR2 levels are 2 to 3 times greater compared to those of wild-type (WT) mice [232]. Also, protein kinase A (PKA)- or calcium/calmodulin-dependent protein kinase II (CaMKII)-mediated hyperphosphorylation and S-nitrosylation of RyR2 have been demonstrated in dystrophic cardiomyocytes and lead to dissociation from the stabilizing-protein calstabin2 and subsequent increased release of calcium from the SR [72,244,245,246]. This is supported by studies demonstrating that agents blocking RyR2 phosphorylation result in the normalization of cytosolic calcium levels as well as improvement in cardiac pathology and arrythmias [245,246,247,248]. SERCA2a activity is significantly reduced in the C57BL/10 *mdx* dystrophic hearts due to lower SERCA2a expression and impaired regulation by inhibitory proteins phospholamban (PLN) and sarcolipin [232,247,248]. Phosphorylation of PLN via PKA or CaMKII is mostly regulated by β-adrenergic signaling [249,250]. Inositol trisphosphate (IP_3_) receptors, which are calcium-release channels on the SR, are activated by the downstream product of phospholipase C (PLC) [236]. PLC inhibitors have demonstrated normalization of intracellular calcium levels in C57BL/10 *mdx* cardiomyocytes to WT levels [251]. It should be noted that calcium-dependent phospholipase A_2_ (PLA_2_) can also be activated when dystrophic cell membranes permit excessive calcium influx, thus resulting in destabilization of muscle membrane structure, at least in dystrophic myofibers [20,252,253].

Sodium ions can also affect mitochondrial–cytoplasmic calcium exchange through the Na^+^-Ca^2+^ exchanger (NCX), which drives calcium out of the mitochondria in exchange for sodium when cytosolic sodium begins to accumulate as a result of dystrophin deficiency-induced microtears. A significant increase in cytosolic sodium levels [254] and elevated levels of NCLX (Na^+^/Li^+^/Ca^2+^) have been observed in cardiomyocytes of C57BL/10 *mdx* mice [255]. To decrease intracellular sodium and calcium overload caused by pathologic reversal of the NCX, a potent and selective sodium-proton exchanger isoform 1 (NHE-1) inhibitor known as rimeporide has been in development [256,257]. NHE-1 is a membrane transporter responsible for catalyzing the electroneutral counter transport of sodium and hydrogen ions through the plasma membrane [256]. Interestingly, when rimeporide was tested on *mdx* mice, protective effects against inflammation and accumulation of fibrosis were observed in the heart, but additional work in this field to identify potential side effects and dose–response curves is still required [256]. Additionally, previous work in gastrocnemius and longissimus dorsi muscles from 5- to 6-month-old C57BL/10 *mdx* mice demonstrated elevations in Na^+^/K^+^ ATPase content [258]. Albeit not in the heart, this further suggests that sodium regulation is abnormal in dystrophic muscle and potentially precedes cell death via abnormal regulation of the cell volume [258] (Figure 2).

Alongside perturbed Na^+^ regulation in dystrophic skeletal and cardiac muscle, the efficiency of cardiac K^+^ ion transport and total cardiac K^+^ content in 8-week-old C57BL/10 *mdx* mice is decreased [259]. This is important because impairments to K^+^ homeostasis have been implicated in the manifestation of dilated cardiomyopathy observed across several myopathies, including DMD [259,260]. Given that several calcium-dependent processes are impacted by dystrophin deficiency, mitochondrial K^+^ channels represent a prospective therapeutic target due to their involvement in mitochondrial large-conductance Ca^2+^-dependent K^+^ channels (mitoBK_Ca_) [259]. Interestingly, it has been observed that the chronic administration of the benzimidazole derivative NS1619 (an activator of mitoBK_Ca_) improved K^+^ transport and content in cardiac mitochondria of 8-week-old C57BL/10 *mdx* mice while simultaneously alleviating lipid peroxidation, reducing cardiac fibrosis, and normalizing the cardiac mitochondrial ultrastructure [259]. This same group also tested a metabolic modulator known as uridine, which is thought to activate the mitochondrial ATP-sensitive K^+^ channel through actions of the metabolite of uridine—UDP (uridine-5-diphosphate) [259,261]. It was revealed that daily administration of uridine for four weeks improved K^+^ transport and content in cardiac mitochondria of 8-week-old C57BL/10 *mdx* mice, akin to NS1619, which was accompanied by elevations to the cardiac mitochondrial number and increases in the calcium retention capacity [259,261,262]. Collectively, these findings highlight opportunities for further investigation into the potential role of ion (e.g., Na^+^, Ca^2+^, Li^+^, K^+^, Cl^−^, and H^+^) dysregulation as a prospective driver of mitochondrial dysfunction.

The disruption of ions could theoretically induce mitochondrial dysfunction. While the relationship between sodium and mitochondrial dysfunction in DMD has not been examined in detail, the link to calcium stress has been investigated extensively. This is an important relationship to consider, given that cardiac mitochondria do not typically act as a significant dynamic buffer of cytosolic calcium in healthy hearts; however, prolonged elevations of intracellular calcium substantially enhance mitochondrial calcium uptake [263,264]. Recently, increased calcium uptake has been observed in the mitochondria of C57BL/10 *mdx* cardiac muscle, likely due to enhanced expression of the channel-forming MCU subunit and reduced expression of the dominant-negative MCUb subunit [255]. While the precise mechanisms regulating excess mitochondrial calcium uptake require more investigation, current theories posit that calcium overload triggers cell death through mitochondrial permeability transition pore activity in conjunction with impaired oxidative phosphorylation and elevated ROS production, as described below.

### 3.3. Mitochondrial Permeability Transition Pore and Apoptosis

Mitochondrial permeability transition pore (PTP) formation is an event that fuses the inner (IMM) and outer mitochondrial membrane (OMM) while permeabilizing the inner membrane. Recent evidence supports a model whereby ATP synthase components dimerize to form the mPTP in response to reactive oxygen species and excess calcium stress [265,266], which then leads to depolarization, a loss of ATP synthesis, and rapid mitochondrial calcium release to the cytosol through an event known as permeability transition (PT). Following the subsequent release of cytochrome *c*, caspases 9/3 are activated, which activate proteases that contribute to apoptosis [267] (Figure 1). There is also evidence that the ADP/ATP antiporter, adenine nucleotide translocase (ANT), may be involved in PTP given that sarcoglycan-deficient mice were shown to develop PTP in an ANT-dependent manner [268].

Myotubes cultured from C57BL/10 *mdx* demonstrate ~10-fold higher calcium in the matrix compared to the cytosol [269]. Calcium-induced PT was elevated following ischemia–reperfusion in young C57BL/10 *mdx* hearts and was accompanied by the release of proapoptotic factors (including cytochrome *c* into the cytosolic fraction), as well as enhanced activities of caspases 9/3 prior to the onset of cardiac dysfunction [70,71] (Figure 1). In LV of 4-week-old D2.*mdx* mice, no differences in calcium-induced PT or caspases 9/3 activities were observed [270] in contrast to increases in both measures seen in certain skeletal muscles [271]. As there were no reductions in the ejection fraction at this young age, the study concluded that mitochondria do not demonstrate PT in LV in early-stage disease. A separate study conducted on 7-month-old C57BL/10 *mdx* mice using mitochondria isolated from whole hearts demonstrated a greater propensity for calcium-induced PT [272]. Collectively, these findings suggest that introducing physiological stressors such as ischemia–reperfusion or assessing more advanced disease states may be required to reveal an underlying mitochondrial propensity for PT, although the degree to which PT exists across specific regions of the heart is unknown. The contributions of PT to cardiomyopathy at specific stages of disease require further research.

It has been proposed that different analytical interpretations can be associated with the specific technique that is used to determine and quantify PT. Techniques include quantifying the concentration of calcium required to trigger its opening via bolus titrations in a permeabilized fiber system versus the elapsed time that is required to trigger this opening in response to a single bolus of calcium. To this end, differing conclusions from studies have led to the belief that disease effects on time may not be mirrored by changes in the total calcium uptake by PT (proposed and reviewed in [14]). In one such study, both approaches were compared and demonstrated no change in total calcium uptake despite a lower time for calcium to trigger PT in C57BL/10 *mdx* mice [71]. This finding implies that a greater propensity for calcium-induced PT may be due to greater velocities of calcium uptake independent of potential differences in total uptake. Such methodological considerations could be considered when assessing PT in relation to cardiomyopathy in *mdx* models.

Although this has not been studied extensively in the heart, pharmacological inhibitors of mPTP in skeletal muscle have shown promise by improving indices of muscle function in C57BL/10 *mdx* mice [14,273,274]. Whether this can be attributed to the inhibitory effects on PT itself or the indirect benefit of immunosuppression is still a contentious topic. This is important given that immunosuppressive glucocorticoids have been known to suppress PT in C57BL/10 *mdx* skeletal muscle [102].

Overall, the specific importance and role of PT-induced cell death in dystrophic cardiomyopathy is still not fully understood and requires more research. Insight may be gained by considering the effects of mPTP inhibitors in the skeletal muscle of *mdx* mouse models. For example, Debio 025 (D-MeAla^3^EtVal^4^-cyclosporin), an inhibitor of the mPTP regulator cyclophilin, showed beneficial roles in locomotor and respiratory muscles of 3-week-old *mdx*^5Cv^ mice [275,276] but had no effect on echocardiograph assessments of heart function or fibrosis in 18-week-old D2.*mdx* mice [75]. Alisporivir, another cyclophilin inhibitor, is a cyclosporin A derivative that desensitizes the PTP without inhibiting calcineurin [277]. Albeit not in the heart, treatment of alisporivir in primary cultures obtained from muscle biopsies of DMD patients demonstrated that this compound could restore the maximal respiratory capacity in dystrophic muscle cells without interfering with basal oxygen consumption, therefore restoring physiological respiratory reserve [277]. In addition, treatment with alisporivir also recovered respiratory function, which matched improvements to muscle ultrastructure and survival in the *sapje* zebrafish model of DMD [277]. TR001, which is a metabolically stable triazole analog and inhibitor of mPTP, improved markers of motility 6 days post-fertilization when administered in *sapje* zebrafish [278]. In addition to improving muscle structure and function recovery, as well as mitochondrial respiration and survival in these zebrafish, TR001 also improved respiration in skeletal muscle-derived myoblasts and myotubes from DMD patients [278]. The promise and efficacy of mPTP inhibitor treatment modalities on dystrophic locomotor and respiratory muscle in various pre-clinical DMD models supports the need for these compounds to be tested on cardiac tissue and/or cardiomyocytes.

**Figure 1 cells-13-01168-f001:**
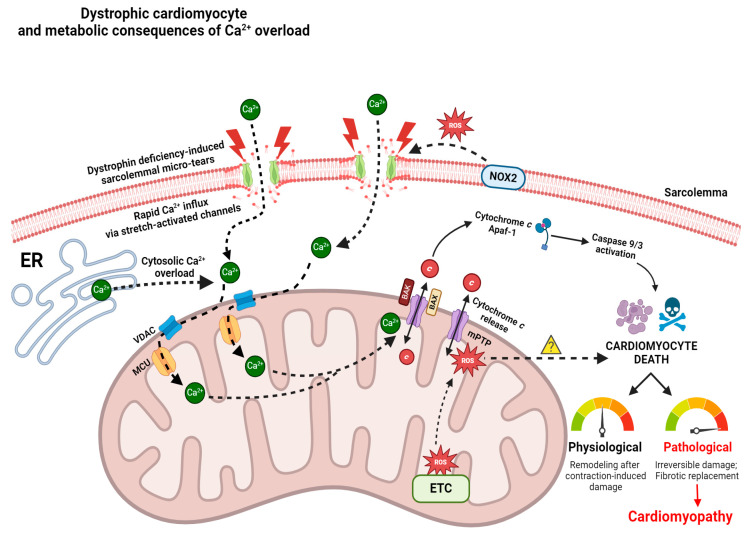
Role of pathological Ca^2+^ overload in perpetuating cardiomyocyte death, and mPTP as a prospective therapeutic target for attenuating cardiomyopathic symptoms in DMD. Dystrophin deficiency-induced sarcolemmal tears lead to rapid cytosolic Ca^2+^ influx via stretch-activated channels (SACs) along the membrane [48,232,233]. NOX2-derived oxidative damage to SACs further damages these channels, thus forcing additional Ca^2+^ to enter the cytosol [42,237]. During cytosolic Ca^2+^ overload, the mitochondrion takes up excess Ca^2+^ via voltage-dependent anion channel (VDAC) and mitochondrial calcium uniporter (MCU) channels along the OMM and IMM, respectively. Subsequent mitochondrial Ca^2+^ overload results in mPTP opening, which expels cytochrome *c* into the cytosol [70]. It is also hypothesized that complex I and complex III-derived reactive oxygen species (ROS) damage stimulates mPTP opening (insufficient data); however, further research is required to characterize the presence of complex III-derived ROS in *mdx* models [70,270]. It is also presently unknown if mitochondrial ROS directly triggers cardiomyocyte death in dystrophic hearts (denoted by yellow question mark). Cytochrome *c* release from the OMM interacts with the apoptosome-containing adaptor Apaf-1 and other initiator caspases (9/3) [71,267], which induce cardiomyocyte death downstream. In healthy physiological systems, regulated control of cardiomyocyte death is required for remodeling events following contraction-induced damage; however, in pathological states, excessive cardiomyocyte death leads to irreversible damage and fibro-fatty tissue replacement, also known as cardiomyopathy. VDAC—voltage-dependent anion channel; MCU—mitochondrial calcium uniporter; ER—endoplasmic reticulum; ETC—electron transport chain; mPTP—mitochondrial permeability transition pore; Apaf-1—Apoptotic protease activating factor 1; NOX2—NADPH oxidase 2; Ca^2+^—calcium ion; ROS—reactive oxygen species; BAX—Bcl-2-associated X protein; BAK—Bcl-2 antagonist killer; *c*—cytochrome *c*. Created on BioRender.com accessed on 1 July 2024.

### 3.4. Mitochondrial Biogenesis

Mitochondrial biogenesis is regulated, in part, through the sequelae of transcriptional regulators that activate or repress promoters for genes encoding over 1500 mitochondrial proteins. Time-dependent decrements to peroxisome proliferator-activated receptor γ coactivator 1-α (PGC-1α) expression—transcriptional co-activator regulating numerous transcription factors—were identified in TA muscle of C57BL/10.*mdx* mice [279]. This finding corresponded to reduced expression of target genes. Several seminal papers have demonstrated that regulation of PGC-1α levels in the skeletal muscle of mdx mice may represent a potential avenue for reducing dystrophic muscle pathology given the beneficial effects that have been reported, including improvements to muscle histology, functional performance, and fatigue resistance [280,281,282]. Although studies have explored AMPK phosphorylation, which regulates PGC-1α in response to a variety of stimuli, and PGC-1α activation in TA muscle of C57BL/10.*mdx* mice using metformin to demonstrate upregulations in mitochondrial content [283], heart-specific data remain limited and represent a future avenue for research.

Contrary to these findings, a separate study assessing mitochondrial biogenesis in pre-necrotic 2-week-old C57BL/10.*mdx* mice revealed no differences in PGC-1α protein content in the quadriceps muscle [284]. These conflicting data suggest that the regulation of mitochondrial biogenesis in mdx mice may be complex and age- and muscle-specific, but the transient states of activity in this co-activator, as well as other transcriptional processes, requires careful consideration of timepoint selection when relating mitochondrial adaptations to disease processes.

Changes in transcription factors and co-activators that govern mitochondrial biogenesis do not always predict mitochondrial functions. This is due to the complexity by which mitochondrial transporters and enzymatic pathways are regulated post-translationally and in response to other factors, such as the morphology and structure of the mitochondrial membranes. Furthermore, transcriptional pathways are often induced in response to a stressor that signals a demand for altering mitochondrial content, as can occur under conditions of insufficient energy supply. In this way, simply measuring markers of transcription factors does not necessarily predict functional outcomes of mitochondria. As an example, the mitochondrial permeability transition pore inhibitor Alisporivir increased certain measures of mitochondrial respiration despite lowering the mRNA of PGC-1α [285]. While speculative, these findings may suggest that the improved mitochondrial function may have alleviated a stress signal that was driving the demand for mitochondrial biogenesis. In this way, contextualizing markers of mitochondrial biogenic pathways in relation to mitochondrial function, and ideally mitochondrial content markers, can provide additional insight into the manner in which mitochondria do or do not contribute to myopathy.

Collectively, these data suggest that understanding the role of mitochondrial biogenesis in the context of dystrophinopathy is still incomplete and requires further research, with specific considerations that should be made for the time-point (age), muscle-type, and dystrophin-deficient model.

### 3.5. Oxidative Phosphorylation and Substrate Catabolism

Mitochondrial ATP synthesis occurs predominantly through oxidative phosphorylation within the electron transport chain (ETC), a process vital for the energy-demanding heart. In healthy mitochondria, this process depends upon the governance of membrane potential across the IMM and is stimulated by temporary elevations to mitochondrial matrix calcium caused by contraction [286,287]. The early identification of mitochondrial deficits in muscular dystrophies using direct assessments of respirometry in isolated mitochondria in people with DMD was reported in 1967, albeit in skeletal muscle [288]. Other reports around this time focused on skeletal muscle animal models of undefined muscular dystrophies [229,289,290,291] prior to the identification of the *mdx* mouse [69]. Numerous factors need to be considered when assessing the regulation of mitochondrial oxidative phosphorylation in muscle, including phenotype severity (due to the progressive nature of dystrophin deficiency) and the fact that not all mitochondrial protein markers uniformly change across a spectrum of muscle types, models, or age ranges [14]. This is important to note because mitochondrial protein content should not always be interpreted as being reflective of mitochondrial function (discussed in [14]).

While no study has performed functional assessments of cardiac mitochondrial oxidative phosphorylation in people with DMD, one study found that iPSC-derived cardiomyocytes from adults with DMD had significant reductions in oxygen consumption coupled with ATP synthesis [292]. Attenuations in the phosphocreatine (PCr) energy system indicated by lower PCr concentrations were also observed in these cells [292]. In the C57BL/10 *mdx* mouse, studies have shown significant reductions in the ratio of cytoplasmic PCr to ATP ratio, which reflects a compromised ability to maintain energy homeostasis prior to the development of cardiac fibrosis [293]. This observation is consistent with many reports of attenuated mitochondrial respiration in the dystrophin-deficient heart [14].

Of interest, mitochondria can also export phosphocreatine as an alternative to ATP [294,295]. Regulated by mitochondrial creatine kinase (mtCK), this model posits that PCr is recycled back to ATP by local ATP-hydrolyzing proteins through cytoplasmic creatine kinases (Figure 2). As PCr diffuses at a rate several fold faster than ATP, with the creatine product returning to mitochondria ~2000 fold faster than ADP, this fast phosphate shuttling mechanism may be advantageous to maintaining energy homeostasis in cells that can experience very high rates of demand for ATP, such as the heart [295,296,297]. One study found that this faster creatine-dependent phosphate shuttling mechanism was more attenuated than the direct ATP export system when assessed with respirometry in permeabilized muscle fibers from the LV of 4-week-old D2.*mdx* mice [270], which has also been reported in skeletal muscle [271,298]. Such “creatine insensitivity” occurring early in the disease was suggested to be a potential unique mechanism preceding the eventual development of cardiomyopathy.

**Figure 2 cells-13-01168-f002:**
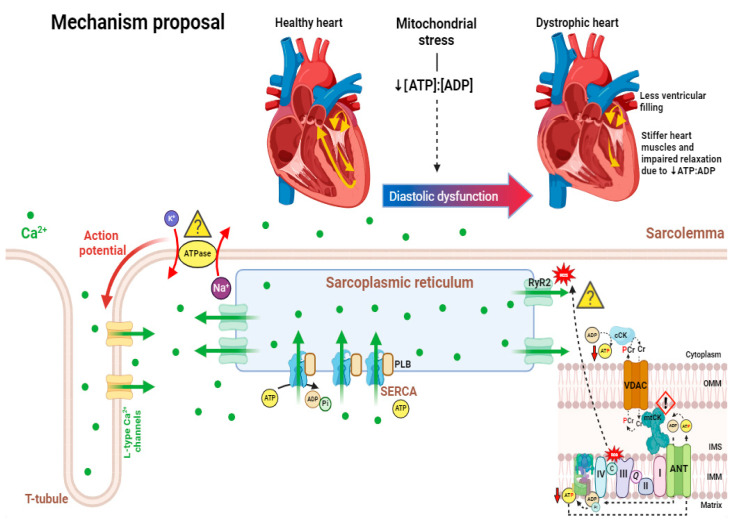
Proposed mechanism demonstrating how dystrophin-deficiency-induced cardiac mitochondrial dysfunction contributes to diastolic dysfunction. As an outcome of mitochondrial dysfunction, left ventricular [ATP]:[ADP] is decreased, while mitochondrial reactive oxygen species (mH_2_O_2_) emission is elevated [255,270]. Yellow question marks denote pathways where ATP insufficiency and elevated mH_2_O_2_ provision may contribute to impaired muscle function, with independent hypotheses broadly reflecting extensive previous literature. Pathways include Na^+^/K^+^ ATPases along the t-tubule (which require ATP to maintain action potential charge balance and membrane excitability) [258], ATP-dependent SERCA (which requires ATP re-uptake of excess cytosolic Ca^2+^ to allow for cardiac relaxation) [232,247,248,258], and ROS-induced RyR2 oxidative stress (which may aberrantly release Ca^2+^ from its SERCA storage site) [245,246,247,248]. Previous work by the authors of [270] also determined that loss of phosphate shuttling by creatine in the LV of D2.*mdx* may also contribute to lower ATP export out of the mitochondria (possible impairment to mtCK denoted by exclamation mark symbol). Potentially due to elevated mitochondrial dysfunction and consequent decreased [ATP]:[ADP], the corresponding stiffer heart muscles are hypothesized to precede diastolic dysfunction exhibited in various late-stage dystrophic models. PLB—phospholamban; SERCA—sarcoplasmic/endoplasmic reticulum Ca^2^⁺-ATPase; RyR—ryanodine receptor; PCr—phosphocreatine; Cr—creatine; Pi—inorganic phosphate; cCK—cytosolic creatine kinase; mtCK—mitochondrial creatine kinase; VDAC—voltage-dependent anion channel; ANT—adenine nucleotide translocase; IMM—inner mitochondrial membrane; IMS—intermembrane space; OMM—outer mitochondrial membrane; I—complex I; II—complex II; Q—coenzyme Q/ubiquinone; III—complex III; *c*—cytochrome *c*; IV—complex IV; K^+^—potassium ion; Na^+^—sodium ion; Ca^2+^—calcium ion; ATP—adenosine triphosphate; ADP—adenosine diphosphate; ROS—reactive oxygen species. Created on BioRender.com (accessed on 1 July 2024).

Additional work is required to address the fates of glucose and other substrates being utilized in dystrophic cardiac tissue. When assessing new therapeutic avenues, careful consideration should be given to designing experiments that consider the role of mitochondrial oxidative phosphorylation as part of an integrated perspective on substrate fates [14].

A variety of other metabolic changes occur in the dystrophic heart. For example, a shift in metabolism from fatty acid to carbohydrate oxidation was observed in vivo in C57BL/10 *mdx* hearts prior to cardiomyopathy, suggesting that altered mitochondrial metabolism is one of the first metabolic changes to occur [299]. In ex vivo perfused C57BL/10 *mdx* hearts, a shift in substrate selection from long chain fatty acids (LCFA) to carbohydrate (glucose) oxidation in C57BL/10 *mdx* hearts was demonstrated by a ~30% lower oleate flux ratio, ~120% higher pyruvate decarboxylation flux ratio, and ~80% increased glycolysis rate [299]. This shift is supported by evidence from cardiac positron emission tomography (PET) studies in DMD patients using ^18^F-deoxyglucose or a radioiodinated branched fatty acid [300,301,302,303]. On the note of carbohydrate oxidation, glucose transporter type 4 (GLUT4) is the main transporter of glucose in cardiomyocytes [304,305] and glucose transport into muscle cells in response to insulin or contraction occurs via translocation of GLUT4 from the cytoplasm to the sarcolemma/T-tubules [305,306]. GLUT4 was observed to be abnormally localized in the sarcolemmal membrane and a trend toward elevated GLUT4 expression was observed in skeletal muscle of GRMD, but not in the LV [307]. Primary cardiac insulin resistance was also observed in the LV of GRMD [308]. Together, this data suggests that glucose/carbohydrate metabolism can potentially be used as a preclinical biomarker of the dystrophic phenotype in skeletal muscle of GRMD dogs, but whether this can be implemented in cardiac tissue remains to be determined [307].

Interventions that correct the substrate imbalance may lead to improvements or recovery of the cardiac contractile dysfunction (reviewed in [309]). Improved contractile performance was observed in C57BL/10 *mdx* hearts perfused with multiple substrates, including carbohydrates (glucose, lactate, and pyruvate) and a LCFA (oleate), in comparison to C57BL/10 *mdx* hearts with glucose as the sole substrate [70,310]. Another potential explanation for substrate shift is the p38 mitogen-activated protein kinase (MAPK) pathway, which is involved through the nuclear receptor peroxisome proliferator-activated receptor (PPARα), a transcriptional regulator of fatty acid oxidation enzyme expression [299]. A 2-fold decrease in p38 MAPK phosphorylation may account for the decrease in LCFA oxidation observed in the C57BL/10 *mdx* heart [70]. This emphasizes the importance of increased and balanced substrate supply to maintain adequate cellular energy in the C57BL/10 *mdx* heart.

It is also possible that certain metabolic stress responses are compensatory in nature. For instance, a shift from fat to carbohydrate oxidation could be beneficial given carbohydrate oxidation produces more adenosine triphosphate (ATP) for a given amount of oxygen consumption [311,312]. However, in DMD, competition for glucose, fatty acids, and proteins may exist for uses unrelated to bioenergetics, such as for membrane repair, as reviewed elsewhere [14]. For example, the degree to which dystrophic muscles downregulate mitochondrial oxidative phosphorylation to favor non-ATP related uses of glucose or fatty acids for structural roles as never been addressed. Such metabolic reprogramming occurs in other models, such as cancer [313], and represents a new avenue for research in DMD. Interestingly, greater levels of fatty acid biosynthesis enzymes that convert fatty acid constituents to acetyl-CoA for the tricarboxylic (TCA) cycle were observed in skeletal muscle from people with DMD [314]. Despite this finding, decreases in TCA pool size and lower aconitase activity were found in C57BL/10 *mdx* hearts [299,315] which could limit oxidative phosphorylation. Likewise, lower mitochondrial respiratory sensitivity to ADP, reduced mitochondrial creatine metabolism, and attenuated complex I activity were reported in LV of 4-week-old D2.*mdx* mice [270]. Some of these metabolic changes [270,299,315] preceded the development of overt cardiac dysfunction, suggesting metabolic stress could be an initial event in the disorder.

The relationship of such mitochondrial reprogramming and/or deficiencies to impaired nitric oxide (NO)/cyclic guanosine monophosphate (cGMP) signaling is relatively understudied in DMD, but several reports supporting continued investigation in this area are warranted. NO/cGMP is believed to regulate nuclear gene expression supporting mitochondrial biogenesis [316]. Increased expression of the atrial natriuretic factor (*anf*) gene, an activator of the NO/cGMP signaling pathway and marker of cardiac remodeling, was observed in C57BL/10 *mdx* hearts at 10–12 weeks of age [299]. Increased levels of the alpha unit of soluble guanylate cyclase (*sgcα_1_*) gene were observed in hearts of 25-week-old mice, a gene which typically negatively correlates with NO/cGMP [317]. These results suggest that an increase in *anf* expression may compensate for a defective NO/cGMP pathway in young mice, but this adaptive mechanism appears to diminish with age [299,317]. However, despite lower nNOS protein content, a study examining skeletal muscle in C57BL/10 *mdx* reported that nitrate supplementation (an approach used to increase NO) did not rescue mitochondrial oxidative phosphorylation, increased mitochondrial peroxynitrite, and promoted muscle damage [318]. Future research could consider whether nitrate supplementation is not effective in *mdx* mice given NO signaling is highly compartmentalized within cells while supplementation cannot mimic this compartmentalized delivery of NO and activation of specific signaling pathways [319]. Collectively, these data suggest that a defect in the NO/cGMP signaling pathway potentially contributes to the metabolic abnormalities in the dystrophic heart [70,299] but negative effects of nitrate supplementation in skeletal muscle of *mdx* mice identify an incomplete understanding of this pathway’s role in contributing to mitochondrial dysfunction in DMD.

Collectively, the regulation of mitochondrial substrate oxidation is altered in the heart during DMD. However, the degree to which such stress responses contribute to cardiomyopathy or serve as an intentional reprogramming to permit alternative fates of glucose or fatty acid substrates requires considerable attention given the latter possibility has rarely been considered (proposed in [14]).

### 3.6. Reactive Oxygen Species (ROS) and Mitochondrial H_2_O_2_ Emission

Many studies measuring ROS in *mdx* models used relatively non-specific fluorophores that do not permit definitive conclusions regarding the precise subcellular origin or type of ROS assessed [14]. In this regard, very few studies have used methodologies that isolated the source of ROS from mitochondria or other sources.

Increased pyruvate-stimulated, complex I-supported mH_2_O_2_ emission was observed during attenuated oxidative phosphorylation in permeabilized LV cardiac muscle fibres of dystrophin-deficient mice due to a reduced ability of ADP to suppress mH_2_O_2_, as normally occurs during oxidative phosphorylation [287]. As creatine normally accelerates mitochondrial ADP/ATP cycling [295], and has been shown to enhance ADP’s ability to attenuate mH_2_O_2_ [320], the additional finding that mitochondrial ADP attenuation of mH_2_O_2_ was no longer sensitive in D2.*mdx* LV fibres suggests that mitochondrial creatine kinase may represent a unique mechanism of mitochondrial dysfunction in dystrophin deficient hearts [270]. The precise mechanisms by which this elevation in mH_2_O_2_/O_2_ may contribute to cardiac dysfunction remains unknown, particularly given there were no changes in the glutathione equilibrium [270] or the ability of mitochondria to scavenge H_2_O_2_ in C57BL/10 *mdx* hearts [71], but such elevations in mH_2_O_2_/O_2_ occurred during a time of apparent cardiac compensations as mice showed slight elevations in ejection fraction albeit before overt cardiac dysfunction [270].

Similar observations were made in isolated mitochondria from mixed hearts (rather than a specific chamber) in the C57BL/10.*mdx* mouse also at 4 weeks of age [255] as well as skeletal muscle of D2.*mdx* mice [52,271,298]. While various antioxidants that do not target mitochondria *per se* have been tested in C57BL/10 *mdx* mice for their potential cardioprotective properties, and have demonstrated discordant results [321], to our knowledge, there are no studies assessing the effects of mitochondrial-targeted antioxidants on mitochondrial bioenergetics and cardiac function in *mdx* models.

The precise signaling mechanisms downstream of mH_2_O_2_ that may mediate cardiac dysfunction in dystrophin deficient hearts have not been explored extensively in *mdx* mice or DMD patients. Understanding how mitochondrial-derived ROS contributes to cardiac dysfunction with consideration of ROS-sensitive targets mediating cell membrane injury, metabolic dysfunction, calcium dysregulation, and cardiomyocyte degeneration could guide the development of ROS-specific targeted therapies.

The superoxide-generating enzyme Nicotinamide Adenine Dinucleotide Phosphate (NADPH) is a major source of non-mitochondrial ROS in skeletal and cardiac muscle [322,323], especially during mechanical stress and at early stages of disorder progression [72,254] (Figure 1). NOX2-mediated ROS generation is significantly increased in C57BL/10 *mdx* hearts [232,322,323,324], which has been attributed to altered microtubule association with NOX2 [324,325]. Inhibiting NOX2-derived ROS [326,327] or using the non-specific ROS scavenger N-acetylcysteine (NAC) [232,322] restored calcium homeostasis in dystrophin-deficient muscle. NOX2 inhibition with the drug apocynin improved single sarcomeric shortening in isolated cardiomyocytes [323] while NAC improved fractional shortening [232]. However, the therapeutic potential of NAC is uncertain given it also reduces muscle weights in C57BL10/*mdx* mice [328] although the degree to which this occurred because of attenuated superoxide is not clear.

### 3.7. Altered Mitochondrial Autophagy (Mitophagy)

In healthy tissue, defective mitochondria can be removed through mitochondrial autophagy, also known as mitophagy, in a process intended to uphold quality control of the healthy mitochondrial pool. This process mitigates damage that may be caused by dysfunctional mitochondria [329] to maintain energy metabolism stability [330,331]. Mitophagy is regulated by PTEN-induced kinase 1 (PINK1) and Parkinson juvenile disease protein 2, (PARKIN) (reviewed in [298]). In a damaged mitochondrion, PINK1 is not degraded and is able to phosphorylate mitofusin 2 (Mfn2), which subsequently signals PARKIN to tag the mitochondria for degradation [329,332]. Important mitophagy-related genes, including *PINK1*, *PARK2*, and *BNIP3* were decreased in DMD patients (dystrophic quadriceps) and dystrophin-deficient rodent models (12-month-old C57BL/10 *mdx* mice) [332,333]. One study showed a relationship between increased mitophagy and improved diaphragm force and histopathology, as well as reduced mitochondrial permeability transition pore, when activating AMPK with AICAR [334], but further research is required to develop mitophagy-specific therapeutics given AMPK regulates other processes that could contribute to the improved phenotype. Collectively, further research is required to elucidate if PINK/PARKIN pathways represent potential therapeutic targets in the dystrophic heart.

Given that loss of PINK1 increases the vulnerability of the heart to IR-injury, it appears to possess some cardioprotective properties [332]. A study by [335] revealed that the co-localization of LC3 dots with fragmented mitochondria was significantly increased in hearts of 22-week-old C57BL/10 *mdx* mice compared to healthy controls, indicating that the elimination of damaged mitochondria via mitophagy was impaired and subsequent accumulation of damaged mitochondria persisted. Other studies in skeletal muscle from C57BL/10 *mdx* mice and DMD patients demonstrated significantly reduced levels of autophagy [336], suggesting that the suppression of autophagy may be muscle-type dependent. Further, mitophagy is a dynamic process and therefore, changes to protein content and gene expression of common mitophagic markers may not always be proportional to mitophagic activity and/or flux. It is essential to understand phenotype severity, muscle-type, and age when considering mitophagy as a therapeutic target. The majority of research to date investigating the role of mitophagy in dystrophin deficient models has been conducted in skeletal muscle, thus, advancements in cardiac muscle remain to be addressed.

### 3.8. Altered Mitochondrial Content and Structure

iPSC-cardiomyocytes from DMD patients possess increased mitochondria with abnormal morphologies [292]. Degeneration of mitochondrial structure (swelling and loss of cristae) has been identified in dystrophic cardiomyocytes from hearts of 1-month-old and 3–4-month-old *mdx* mice prior to disorder onset [254]. Furthermore, a significant increase in structurally abnormal mitochondria were observed in hearts of 12-month and older C57BL/10 *mdx* mice after developing DCM [332] which has also been reported as early as 4 weeks of age in this model [255]. Various mitochondrial proteins contents have been assessed in *mdx* mouse models with divergent responses observed depending on the pathway, age and model suggesting that altered control of oxidative phosphorylation could occur through post-translational modifications that have yet to be fully identified, and that compensations in the content of certain pathways may occurs [14,71,270].

## 4. Mitochondria as a Potential Therapeutic Target

### 4.1. Targeting Calcium Handling

Given mitochondrial dysfunctions are thought to arise, in part, by elevated calcium uptake, preventing mitochondrial dysfunction could be achieved by improving cytoplasmic handling itself. For example, due to the hypersensitivity to excitation-contraction coupling in cardiomyopathy of DMD patients, targeting RyR2 may offer therapeutic benefit. PKA phosphorylation of RyR2 appears to contribute to increased calcium release from the SR, and therefore, PKA could be a potential therapeutic target (Figure 2). In dystrophic mice, inhibition of PKA-mediated phosphorylation of RyR2 reduced SR calcium leak and in turn prevented cardiomyopathy [337]. Oxidative stress can lead to the nitrosylation of RyR2 and subsequent disassociation of calstabin2 from RyR2, increasing SR calcium leak [245]. Treatment of C57BL/10 *mdx* mice with either NAC to inhibit RyR2 nitrosylation, or the RyR2 stabilizer Rycal, prevented depletion of calstabin2 and subsequent SR calcium leak, aberrant depolarization in cardiomyocytes, and arrhythmias [245].

The stretch-sensitive channel TRPV2 is another calcium channel target to consider, as increased TRPV2 has been documented in the cytoplasmic membrane of *mdx* cardiac and skeletal cells and the sarcolemmal membrane of DMD patients [338]. Treatment of DMD patients with the antiallergy drug Tranilast, which has anti-TRPV2 activity, resulted in reduction of heart failure biomarkers [339].

Enhancing SR calcium uptake through overexpression of SERCA or targeting its inhibitors can be considered in restoring calcium homeostasis. In C57BL/10 *mdx* mice 12 months of age, administration of AVV9-SERCA2a gene therapy significantly improved cardiac electrophysiology [340], whereas in 3-month-old *mdx* mice a similar therapy ameliorated DCM for at least 18 months [341]. Decreasing sarcolipin expression in *mdx* mice restored cardiac SERCA function and calcium cycling, thus preventing cardiomyopathy. Decreased LV internal diameter in diastole as well as decreased fibrotic and necrotic tissue likely contributed to the improved cardiac function [248]. While targeting sarcolipin may be a promising approach, cardiac function worsened in phospholamban (PLN) knockout C57BL/10 *mdx* mice [342].

### 4.2. Mitochondrial Calcium Overload

As described previously, frequent PT activity may contribute to calcium-dependent mitochondrial dysfunction and therefore, prevention of this opening could be beneficial. As discussed in Section 3, inhibition of a key regulator of mPTP, cyclophilin, may impede opening of the pore and subsequent mitochondrial dysfunction [273,275].

### 4.3. Targeting Cellular/Mitochondrial Antioxidant Systems

Since cardiac mH_2_O_2_ is elevated early in DMD [255,270], mitochondrial-targeted antioxidant therapies could provide therapeutic benefit in DMD patients. Several general antioxidants, including MitoQ and SkQ1, are conjugated with the lipophilic cation triphenylphosphonin (TTP) to target the mitochondria [343]. The small ‘SS’ peptides are cell permeable agents that bind to cardiolipin on the IMM and can preserve oxidative phosphorylation, attenuate mH_2_O_2_, and prevent oxidative damage [344,345,346]. Furthermore, SS peptides can inhibit the dissociation of cytochrome *c* from cardiolipin, hindering the activation of cell death pathways [347] (Figure 3). To our knowledge, no publication has investigated the effects of any mitochondrial-ROS lowering compound in models of DMD.

### 4.4. Cardiolipin and Membrane Stability

Cardiolipin, a phospholipid enriched with unsaturated fatty acids located in the IMM, is essential to maintain mitochondrial structure and function. Through its binding to key modulators of energy exchange and subsequent proteolipid complex formation, cardiolipin plays a key role in mitochondrial bioenergetics [295] (Figure 3). This complex can regulate energy exchange, reduce mitochondrial ROS generation, and prevent opening of mPTP [295]. In the presence of oxidative stress, cardiolipin and the key regulator mtCK, lose their binding capacity, leading to dissociation of the proteolipid complex [348,349] (Figure 3). Since mH_2_O_2_ generation is elevated in dystrophin deficient heart [255,270], a cardiolipin targeting agent, such as elamipretide, may be appropriate for targeting impaired mitochondrial bioenergetics (Figure 3). Although data on cardiolipin physiology in *mdx* is limited, the altered mitochondrial cristae structure reported in Section 3.8 serves as a foundation for continuing the investigation on cardiolipin in various dystrophic models.

**Figure 3 cells-13-01168-f003:**
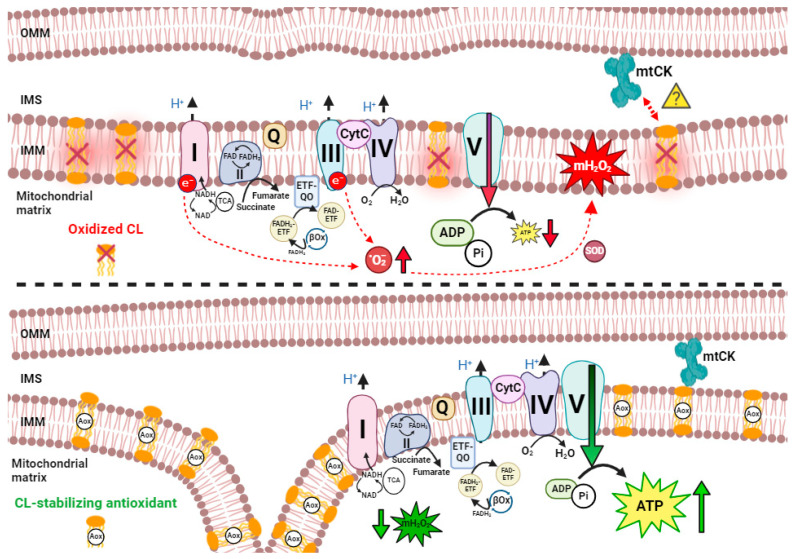
Cardiolipin as a prospective therapeutic target for treating mitochondrial dysfunction in dystrophic cardiac tissue. Since cardiolipin (CL) is highly redox-sensitive (attributed to its high structural content of unsaturated fatty acids), and mitochondrial reactive oxygen species (mH_2_O_2_) provision is elevated in dystrophic cardiac tissue [270], targeting cardiolipin may be beneficial for stabilizing mitochondrial membranes. (**Top panel**) Herein, we propose that when CL is oxidized, electron transport chain (ETC) super-complexes are unevenly distributed, thus resulting in elevated mH_2_O_2_ production and reductions to [ATP]:[ADP] [348,349]. Additionally, mitochondrial creatine kinase (mtCK) may detach from CL (denoted by red arrow; question mark due to insufficient data), while in a healthy physiological state, CL is shown to anchor mtCK [295]. (**Bottom panel**) In the presence of a cardiolipin-stabilizing agent such as elamipretide (denoted by Aox (antioxidant) peptide stabilizing CL), ETC super-complexes can be tethered to each other while CL anchors mtCK. This results in lower mH_2_O_2_ emissions and a higher [ATP]:[ADP] ratio. Both panels demonstrate that electrons derived from the TCA cycle support matrix NADH production (for oxidation by Complex I) and succinate-supported FADH_2_ production internal to CII, as well as matrix FADH_2_ derived from β-oxidation that supports entry to the ETC through ETF and ETF-QO [350]. Additional electron entry sites to the Q junction, including glycerol 3 phosphate dehydrogenase, are described elsewhere [351]. CL—cardiolipin; TCA—tricarboxylic acid cycle; mtCK—mitochondrial creatine kinase; IMM—inner mitochondrial membrane; IMS—intermembrane space; OMM—outer mitochondrial membrane; I—complex I; II—complex II; Q—coenzyme Q/ubiquinone; III—complex III; CytC—cytochrome *c*; IV—complex IV; ATP—adenosine triphosphate; ADP—adenosine diphosphate; Pi—inorganic phosphate; H^+^—hydrogen ion; e^−^—electron; •O2^−^—superoxide; NAD—nicotinamide adenine dinucleotide; FAD—flavin adenine dinucleotide; O_2_—oxygen; H_2_O—water; SOD—superoxide dismutase; ETF—electron transport flavoprotein; Aox—Antioxidant. Created on BioRender.com accessed on 1 July 2024.

Albeit not published in a dystrophin-deficient model, SS-31, or elamipretide, is an agent that has demonstrated improvements to mitochondrial functions including reduction of mitochondrial ROS across several pathologies [352,353,354]. Elamipretide has also shown evidence of direct cardioprotective mechanisms, including the amelioration of apoptosis and fibrosis in preclinical models of heart failure [355,356].

Idebenone, a synthetic analogue of Coenzyme Q_10_ with electron-shuttling activity, can alter respiratory chain activity [357,358], although this may require very high micromolar doses [359,360]. Long-term idebenone therapy improved cardiac diastolic dysfunction, reduced cardiac inflammation and fibrosis, improved running performance, and limited mortality from cardiac pump failure induced by dobutamine stress testing in vivo in *mdx* mice [361]. Early clinical trials in patients with DMD revealed improved cardiac and respiratory function with idebenone treatment [362,363,364]. Specifically, a trend for increased peak systolic radial strain in the LV was seen in these patients [362]. However, no measures of mitochondrial bioenergetics were performed to verify that the drug was acting through a mitochondrial mechanism. As reviewed elsewhere, idebenone has a variety of effects on other organelles, which may not involve the antioxidant properties of the compound [365]. This suggests that the beneficial effects in some studies may not be due to alterations in mitochondrial functions despite claims to this effect. These factors might be consistent with a subsequent clinical trial in people with DMD that was unsuccessful (NCT02814019). As such, no study to date has definitively tested the potential of mitochondrial-targeted therapies on cardiac dysfunction in pre-clinical models of dystrophin deficiency or in clinical studies.

Lastly, the potential for mitochondrial therapies to be combined with the current standard of care (glucocorticoids) or emerging gene-based and vector-mediated therapies requires extensive research to determine the potential for combinatorial treatments.

### 4.5. Perspectives on Mitochondrial Dysfunction vs. Metabolic Reprogramming

To date, there are limited studies that have delineated whether mitochondrial responses to DMD should be considered dysfunctions or instead reflect intentional metabolic reprogramming necessary for survival. This is partially exemplified by the authors of [282], who demonstrated that short-term overexpression of the dominant regulator of oxidative metabolism, PGC-1α, via in vivo plasmid transfection in C57BL/10.*mdx* after the onset of dystrophin deficiency-induced necrosis was beneficial for several mitochondrial and non-mitochondrial functions. Briefly, the team determined that overexpressing PGC-1α was sufficient to restore mitochondrial biomass, increase the mitochondrion’s capacity to buffer Ca^2+^, normalize the susceptibility to PTP opening, and reduce the activity levels of caspase 9/3, which are involved in intrinsic apoptotic signaling pathways [282]. While PGC-1α also regulates non-mitochondrial pathways involved in muscle fatigue resistance, the findings support a rationale to use new targeted approaches that isolate mitochondrial roles in myopathy during DMD. In this way, mitochondrial-targeted therapies may assist in determining the causal relationships that exist between mitochondrial responses to DMD and muscle dysfunction.

It is imperative to understand that while mitochondrial responses to dystrophin deficiency are still “stress” responses in that they correspond to a disease stimulus, the term “dysfunction” should be reserved for instances where the response is causal of a myopathy. As we recently proposed in [14], the term “dysfunction” must be used with care when defining a change in mitochondrial function, given that no investigation has thoroughly determined whether such changes are part of a larger metabolic reprogramming that serves a grander purpose within survival programs yet to be identified (e.g., the re-direction of substrates to non-energy fates, as occurs in other models of growth and repair such as cancer) [313]. Therefore, while this review routinely used the term “mitochondrial dysfunction”, the corresponding literature could be reconsidered through the perspective of “mitochondrial stress responses” until such time that investigations thoroughly establish direct causal linkages between these responses and muscle dysfunction.

Nonetheless, the majority of stress responses in the current literature support the notion that certain mitochondrial responses to dystrophin deficiency are likely causal of myopathy and represent a prospective target for therapy development. Uncovering such avenues for alternative therapeutic interventions that address secondary contributors to pathology in DMD is imperative while gene-targeted and other similar therapies continue to develop and become widely available for clinical use.

## 5. Summary and Future Directions

Since emerging therapies are thus far unable to completely restore the *DMD* gene, there remains a need to develop additional therapies treating secondary contributors to cardiomyopathy. While curative approaches customized to individual mutations are actively pursued, developing new combinatorial therapies targeting a variety of cellular and physiological stressors could complement existing conventional approaches in a manner that benefits the majority of people with DMD. In this regard, the collective evidence warrants the development of new approaches targeting partial dystrophin restoration, inflammation, calcium and other ion dysregulation, membrane tearing, oxidative stress, cytoskeletal disorganization, metabolic stress, and other cellular dysfunctions. Many of these stressors are related to mitochondrial-specific stress responses linked to disrupted energy homeostasis, redox balance, and calcium-induced cell death. The degree to which each mitochondrial stress response contributes to cardiomyopathy in DMD as actual mitochondrial dysfunctions requires careful consideration of the mechanisms of specific mitochondrial targeting compounds, the age and type of animal model used in pre-clinical investigations, and cardiac chamber-specific dysfunctions. The overview provided in this article serves as a basis for guiding experimental design in a way that captures the specific mechanisms by which mitochondria contribute to cardiac dysfunction and histopathology in DMD.

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
