# Peer review of "Cardiomyopathy in Duchenne Muscular Dystrophy and the Potential for Mitochondrial Therapeutics to Improve Treatment Response"

_cells, 2024, doi:10.3390/cells13141168_

Round 1

Reviewer 1 Report

Comments and Suggestions for Authors

The review article is well written and provides a comprehensive update on the progress in the research field. I just have some minor concerns.

1) Overall the article can be more focused and concised. For example, in the current format, the part 3.4 did not clearly discuss the relationship between oxidative phosphorylation and cardiomyopathy in the context of DMD.

2) Discussion in mitochondrial-targeting therapy for DMD maybe can be combined with section 4.

3) Section 5 can be combined with Section 2.

Author Response

REVIEWER 1

GENERAL COMMENT:

The review article is well written and provides a comprehensive update on the progress in the research field. I just have some minor concerns.

Minor:

1) Overall, the article can be more focused and concise. For example, in the current format, the part 3.4 did not clearly discuss the relationship between oxidative phosphorylation and cardiomyopathy in the context of DMD.

Response: Thank you for this feedback. We have changed the title of the manuscript to reflect the broader description of the review on clinical and mechanistic considerations of DMD that allows the reader to better understand the context in which mitochondrial contributions may contribute to the pathology. We have also restructured major sections of the review and removed content, while adding new content based on feedback from the reviewers. The mitochondrial section now stands out as a major section that is positioned after a more organized umbrella description of the broader complexities of disease processes in DMD.

In Section 3.05, lines 1016-1033 serve as a preamble that introduces the history of investigating oxidative phosphorylation in DMD models. Lines 1034-1043 specifically discuss the role of oxidative phosphorylation in cardiomyopathy (or at least dystrophin deficiency-induced cardiomyopathic manifestations such as cardiac fibrosis).

2) Discussion in mitochondrial-targeting therapy for DMD maybe can be combined with section 4.

Response: With Section 5 shortened and reconfigured into Section 2.10, the new Section 4 (Mitochondria as a potential therapeutic target) logically follows Section 3 (Inflammation, calcium dysregulation, and their contribution to mitochondrial dysfunctions in DMD) to improve flow of the overall manuscript.

3) Section 5 can be combined with Section 2.

Response: Section 5 has been considerably shortened and now reconfigured to the end of Section 2 (2.10. Cardiac chamber-specific differences in DMD) as suggested by the Reviewers.

Reviewer 2 Report

Comments and Suggestions for Authors

GENERAL COMMENT:

I have carefully reviewed this manuscript entitled Mitochondria in Duchenne Muscular Dystrophy-induced Cardiomyopathy: A Prospective Therapeutic Target to Improve Treatment Response. This review is submitted in the Topical Collection: Feature Papers in Mitochondria. Mitochondrial dysfunction is a well-established contributor to the pathogenesis of DMD, and ample experimental support has been published on this topic over that last 10-15 years, not only in skeletal muscle but also in the heart. This review could potentially fit the mandate of this topical collection. The manuscript provides a comprehensive review of DMD, it is well written, and well-referenced in general. However, there is not enough emphasis placed on mitochondria and the information is diluted. My recommendation would be to shorten the manuscript significantly and increase the focus on mitochondria.

·      The first 15 pages of the manuscript present a review of 1) DMD epidemiology, genetics and clinical presentation; 2) pre-clinical models of DMD, 3) functional and histological assessment of cardiomyopathy, and mechanisms of cardiomyopathy including a critical analysis of advantages and limitations of assessment modalities. Mitochondria are introduced for the first time in section 3.3 discussing Ca2+ dyregulation and mPTP opening.

·      Similarly, much of section 4 (current interventional target) and 5 (chamber specific cardiomyopathy) contain a lot of information that are not relevant to the topic.

·      In section 6 dealing with therapeutic interventions, the title indicate mitochondria as potential therapeutic target, but in reality this section discusses many non-mitochondrial targets that may have indirect effects on mitochondria. I think this could be misleading to some readers It should be made clear at the beginning, perhaps by classifying mitochondrial modulators based on whether they act directly or indirectly.

Minor:

P2 line 71 : mitochondrial dysfunction has come into focus…in fact it’s been a while….almost 20 years now.

P3 line 134: It may be worth adapting the wording a little bit in this paragraph to ephasize that membrane instability alters ion homeostasis through abberant regulation of sarcolemmal channels. As written emphasis is on tears, which only explain part of the problem.

Figures : I did not see reference to figures inserted in the text. This should be added.

P17-Figure 1: The term mitochondrial stress is vague. The author should clarify what they mean by this. For example what are some key alterations in dystrophin-deficient cardiomyocytes that can reasonably be considered a mitochondrial stressors. The large question mark at the top left of the figure is confusing and has little use. I would suggest removing it and to keep only the small ones in the figure. In fact, these questions marks point to sites where authors believe that impaired ATP provision and enhanced oxidative stress alters EC coupling and contraction. I also notice that none of the figures actually present the creatine-dependent phosphate shuttling system, which appears to be a nexus of early dysfunction in DMD.

P18 line 852-853 : What are the experimental evidence supporting an elevation in fat oxidation in DMD? This statement seems to contradict evidence discussed in the following paragraph (ie shift toward glucose oxidation and away from fat oxidation)

P19 line 895-896: The authors mention that in DMD, competition for glucose, fatty acids, and proteins may exist for uses unrelated to  bioenergetics, such as for membrane repair, as reviewed elsewhere (Bellissimo et al., 2023). I believe this should be expanded a bit. This is potentially interesting and more directly related to mitochondrial/metabolic dysfunction than other aspects that could be removed from the review (see general comment).

P19 line 921-924: It could be worth mentioning that because NO/cGMP signaling is compartmentalized, it is perhaps not surprising that nitrate supplementation, which cannot mimic the compartmentalized delivery of NO and activation of specific signaling pathways, has negative effect.

P20 line 997: Improving mitophagy has shown promise in the diaphragm of C57BL/10 mdx mice (Pauly et al., 2012). This should be rephrased. This study did not show causal relationship. The authors used the AMPK agonist AICAR and showed its induced autophagy and improved mPTP function. Mitophagy was not assessed and the benefits of AICAR could me mediated by other mechanisms as well.

P 21 section 4 on  therapies: This section should be condensed to enhance focus on  mitochondria.  For example, in section 4.1, the main point to make is that mitigation of mitochondrial dysfunction may at least partly underlie the beneficial effects of glucocorticoids. Discussion should focus on this, and less on dosage regimen for example. For other common approaches, (b-blockers, Ace inhibitors etc…) mitochondrial implications are not discussed, this moves the focus of the review away from mitochondria.

Section 5 Chamber specific cardiomyopathy suffers from the same problem. I don’t believe this is relevant for a featured papers in mitochondria.

Section 6: The title indicate mitochondria as potential therapeutic target. I think this could be misleading to some readers because section 6.1 goes on to discuss strategies targeting RyR, TRPV2 and SERCA which are not mitochondrial channels. The discussion includes non-mitochondrial drugs targets that have indirect benefits on mitochondria. I think it should be made clear at the beginning….perhaps by classifying mitochondrial modulators based on whether they act directly or indirectly.

P 29-Figure 2: Not clear why this figure is not referred to earlier in the text. It seems to include many aspects that are discussed  10 pages before.

P30 figure 3: The figure legend proposes a model that is not discussed formally in section 6.4. Figure legends states “we propose that when cardiolipin is oxidized, electron transport chain supercomplexes are unevenly distributed and mitochondrial membranes are abnormally linear, thus resulting in elevated mH2O2 production and reductions to [ATP]:[ADP]”. Are there evidence for altered cardiolipin levels or oxidation in DMD or that supercomplexes are perturbed? The legends seems to elaborates on several hypothetical mechanisms for which there is no experimental support.

P 31 line 1492: Regarding the paragraph on the potential role of mitochondria as a modulator of PMOs, it seems to be quite speculative. The hypothesis is based on a 2022 conference abstract reporting that the mitochondria-targeting peptide elamipretide potentiates dystrophin expression induced by an exon-skipping morpholino in the mdx mouse model. The authors propose an hypothesis whereby mitochondria hone in to injury sites to provide ATP and phospholipids,  which could explain how improving mitochondrial function with elamipretide enhanced PMO’s therapeutic effect. This study is not published two years later and there is no way to assess the strength of supporting evidence.

Author Response

REVIEWER 2

GENERAL COMMENT:

I have carefully reviewed this manuscript entitled Mitochondria in Duchenne Muscular Dystrophy-induced Cardiomyopathy: A Prospective Therapeutic Target to Improve Treatment Response. This review is submitted in the Topical Collection: Feature Papers in Mitochondria. Mitochondrial dysfunction is a well-established contributor to the pathogenesis of DMD, and ample experimental support has been published on this topic over that last 10-15 years, not only in skeletal muscle but also in the heart. This review could potentially fit the mandate of this topical collection. The manuscript provides a comprehensive review of DMD, it is well written, and well-referenced in general. However, there is not enough emphasis placed on mitochondria and the information is diluted. My recommendation would be to shorten the manuscript significantly and increase the focus on mitochondria.

The first 15 pages of the manuscript present a review of 1) DMD epidemiology, genetics and clinical presentation; 2) pre-clinical models of DMD, 3) functional and histological assessment of cardiomyopathy, and mechanisms of cardiomyopathy including a critical analysis of advantages and limitations of assessment modalities. Mitochondria are introduced for the first time in section 3.3 discussing Ca2+ dysregulation and mPTP opening.

Similarly, much of section 4 (current interventional target) and 5 (chamber specific cardiomyopathy) contain a lot of information that are not relevant to the topic.

In section 6 dealing with therapeutic interventions, the title indicate mitochondria as potential therapeutic target, but in reality this section discusses many non-mitochondrial targets that may have indirect effects on mitochondria. I think this could be misleading to some readers It should be made clear at the beginning, perhaps by classifying mitochondrial modulators based on whether they act directly or indirectly.

Response: Thank you for this constructive feedback. It helped us realize the experience for the reader was not what we intended. We also realized that the title does not fully capture the intent of providing a broad overview of the clinical and mechanistic understanding of DMD to allow the reader to better understand how mitochondrial contributions fit in the context of the complexity of the disease. In this regard, we have restructured the review by re-organizing certain sections of the review, eliminating other content, and re-stating the title to a more accurate description of this broader intention. As such, the title is now ‘Cardiomyopathy in Duchenne muscular dystrophy and the potential for mitochondrial therapeutics to improve treatment response’.

Minor:

P2 line 71: mitochondrial dysfunction has come into focus…in fact it’s been a while….almost 20 years now.

Response: The wording has been amended: See lines 70-71.

P3 line 134: It may be worth adapting the wording a little bit in this paragraph to emphasize that membrane instability alters ion homeostasis through aberrant regulation of sarcolemmal channels. As written emphasis is on tears, which only explain part of the problem.

Response: This adjustment has been made to emphasize membrane instability and its role in regulation of ion homeostasis and sarcolemmal channels. See lines 152-157. This concept is briefly presented early on and then revisited throughout Section 3.

Figures: I did not see reference to figures inserted in the text. This should be added.

Response: Figures are cited and bolded in-text throughout the manuscript. The notation used is (Fig. #) in-text.

P17-Figure 1: The term mitochondrial stress is vague. The author should clarify what they mean by this. For example what are some key alterations in dystrophin-deficient cardiomyocytes that can reasonably be considered a mitochondrial stressors. The large question mark at the top left of the figure is confusing and has little use. I would suggest removing it and to keep only the small ones in the figure. In fact, these questions marks point to sites where authors believe that impaired ATP provision and enhanced oxidative stress alters EC coupling and contraction. I also notice that none of the figures actually present the creatine-dependent phosphate shuttling system, which appears to be a nexus of early dysfunction in DMD.

Response: The distinction between ‘mitochondrial stress response’ and ‘mitochondrial dysfunction’ has been defined by us in a newly added Section 4.05 (Perspectives on mitochondrial dysfunction vs. metabolic reprogramming). The creatine-dependent phosphate shuttling system has been added in the bottom right corner of Figure 1 and has been referred to in-text and defined in the Figure caption.

P18 line 852-853: What are the experimental evidence supporting an elevation in fat oxidation in DMD? This statement seems to contradict evidence discussed in the following paragraph (ie shift toward glucose oxidation and away from fat oxidation)

Response: We removed any sentences in Section 3.02 referring to fat oxidation for reasons highlighted by the Reviewer. After careful consideration, we concluded that the sentences were purely speculative/hypothesis-generating and did not add substantial value for this manuscript.

P19 line 895-896: The authors mention that in DMD, competition for glucose, fatty acids, and proteins may exist for uses unrelated to bioenergetics, such as for membrane repair, as reviewed elsewhere (Bellissimo et al., 2023). I believe this should be expanded a bit. This is potentially interesting and more directly related to mitochondrial/metabolic dysfunction than other aspects that could be removed from the review (see general comment).

Response: Thank you for this feedback. We are happy to expand on this exciting possible direction or research in myopathies, DMD included, whereby substrate fates beyond ATP synthesis may also be upregulated to support growth and repair. This is an enormous area of research given the carbon backbones and electrons carried in hydrogen bonds of glucose, fatty acids, and proteins are used for structural purpose (eg. glucose to pentose to ribose for RNA and DNA synthesis, fatty acids for membrane biosynthesis) or reductive power that drives these processes, as well as other critical steps like glutathione/thioredoxin turnover, by NADPH (generated with electrons from glucose in the pentose phosphate pathway and TCA cycle, with the latter being co-driven by fatty acids and amino acids). It is an untapped area for research in DMD and muscle physiology and general which could use its own review. For now, we have included the following statement to refer the reader to a useful review of this concept in the field of cancer metabolism (lines 1133-1136):

‘For example, the degree to which dystrophic muscles downregulate mitochondrial oxidative phosphorylation to favour non-ATP related uses of glucose or fatty acids for structural roles as never been addressed. Such metabolic reprogramming occurs in other models, such as cancer (Stine et al., 2022), and represents a new avenue for research in DMD.’

P19 line 921-924: It could be worth mentioning that because NO/cGMP signaling is compartmentalized, it is perhaps not surprising that nitrate supplementation, which cannot mimic the compartmentalized delivery of NO and activation of specific signaling pathways, has negative effect.

Response: We have added the following statements to clarify the role of NO/cGMP signaling and NO supplementation in dystrophic myopathy. See lines 1162-1170.

‘Future research could consider whether nitrate supplementation is not effective in mdx mice given NO signaling is highly compartmentalized within cells while supplementation cannot mimic this compartmentalized delivery of NO and activation of specific signaling pathways (Lundberg and Weitzberg 2022). Collectively, these data suggest that a defect in the NO/cGMP signaling pathway potentially contributes to the metabolic abnormalities in the dystrophic heart (Khairallah et al., 2007; Burelle et al., 2010) but negative effects of nitrate supplementation in skeletal muscle of mdx mice identify an incomplete understanding of this pathway’s role in contributing to mitochondrial dysfunction in DMD.’

P20 line 997: Improving mitophagy has shown promise in the diaphragm of C57BL/10 mdx mice (Pauly et al., 2012). This should be rephrased. This study did not show causal relationship. The authors used the AMPK agonist AICAR and showed its induced autophagy and improved mPTP function. Mitophagy was not assessed and the benefits of AICAR could me mediated by other mechanisms as well.

Response: We have added additional clarification to the statement exploring the relationship between mdx-induced diaphragm myopathy and mitophagy. Specifically, we now mention that mitophagy-specific therapeutics should be developed given that AMPK has pleotropic regulatory roles and may be involved in more pathways than just mitophagy, which may have been unclear in the previous iteration. See lines 1238-1244.

‘One study showed a relationship between increased mitophagy and improved diaphragm force and histopathology, as well as reduced mitochondrial permeability transition pore, when activating AMPK with AICAR (Pauly et al., 2012), but further research is required to develop mitophagy-specific therapeutics given AMPK regulates other processes that could contribute to the improved phenotype. Collectively, further research is required to elucidate if PINK/PARKIN pathways represent potential therapeutic targets in the dystrophic heart.’

P21 section 4 on therapies: This section should be condensed to enhance focus on mitochondria.  For example, in section 4.1, the main point to make is that mitigation of mitochondrial dysfunction may at least partly underlie the beneficial effects of glucocorticoids. Discussion should focus on this, and less on dosage regimen for example. For other common approaches, (b-blockers, Ace inhibitors etc…) mitochondrial implications are not discussed, this moves the focus of the review away from mitochondria.

Response: This Section has been largely condensed and reconfigured into Section 1.07 (Current standard of care and the need for new therapies). We have moved away from dosing regimens but have elected to discuss first-line therapeutics such as ACE inhibitors/ARBs, Beta-Blockers etc. given that they are routinely administered in several cardiomyopathy models, including DMD.

Section 5 Chamber specific cardiomyopathy suffers from the same problem. I don’t believe this is relevant for a featured papers in mitochondria.

Response: This section has been considerably condensed and reconfigured into a single paragraph in Section 2.10 (Cardiac chamber-specific differences in DMD), ending with a perspective on future chamber-specific mitochondrial research avenues.

Section 6: The title indicate mitochondria as potential therapeutic target. I think this could be misleading to some readers because section 6.1 goes on to discuss strategies targeting RyR, TRPV2 and SERCA which are not mitochondrial channels. The discussion includes non-mitochondrial drugs targets that have indirect benefits on mitochondria. I think it should be made clear at the beginning….perhaps by classifying mitochondrial modulators based on whether they act directly or indirectly.

Response: All of these channels play a role in the regulation of calcium transients in cardiomyocytes and are strategically introduced early in Section 4 to build context regarding calcium stress in DMD. Mitochondria-specific interventions are therefore explicitly introduced within Section 4.02 and beyond.

P29-Figure 2: Not clear why this figure is not referred to earlier in the text. It seems to include many aspects that are discussed 10 pages before.

Response: With the rearrangement and removal of several sections, this Figure can now be found cited as early as Line 790.

P30 figure 3: The figure legend proposes a model that is not discussed formally in section 6.4. Figure legends states “we propose that when cardiolipin is oxidized, electron transport chain supercomplexes are unevenly distributed and mitochondrial membranes are abnormally linear, thus resulting in elevated mH2O2 production and reductions to [ATP]:[ADP]”. Are there evidence for altered cardiolipin levels or oxidation in DMD or that supercomplexes are perturbed? The legends seems to elaborates on several hypothetical mechanisms for which there is no experimental support.

Response: The authors state that some of the mechanisms are intentionally speculative/hypothesis-generating based on previous data (cited in the figure caption itself). The ‘question marks’ within the Figure are intended to specify where gaps in knowledge exist. The Figure has been largely reworked to make hypothetical mechanisms (i.e. mtCK-CL relationship) more clear.

P31 line 1492: Regarding the paragraph on the potential role of mitochondria as a modulator of PMOs, it seems to be quite speculative. The hypothesis is based on a 2022 conference abstract reporting that the mitochondria-targeting peptide elamipretide potentiates dystrophin expression induced by an exon-skipping morpholino in the mdx mouse model. The authors propose a hypothesis whereby mitochondria hone in to injury sites to provide ATP and phospholipids, which could explain how improving mitochondrial function with elamipretide enhanced PMO’s therapeutic effect. This study is not published two years later and there is no way to assess the strength of supporting evidence.

Response: Data from non-dystrophin-deficient models demonstrating the effects of Elamipretide as a potential mitochondria-targeted therapeutic has been kept in for the purposes of building context, however, its speculative function of enhancing PMO’s therapeutic effects has been removed along with other data, as mentioned above.

Reviewer 3 Report

Comments and Suggestions for Authors

This is a very high-quality review demonstrating the various manifestations of cardiac pathology in Duchenne muscular dystrophy. The review focuses on mitochondrial dysfunction characteristic of this hereditary pathology. In general, the authors covered the problem well and emphasized further directions for research. However, I had several recommendations and comments regarding its content.

1. I recommend adding a table that summarizes data on the various manifestations of cardiac pathology in DMD (primarily mitochondrial, which is the subject of this review). The table should contain information about the object (human or model animal) demonstrating these manifestations.  

2. Lines 144-147. It should be noted that a number of studies show the absence of a positive effect of Poloxamer 188 on the development of DMD (like doi: 10.1016/j.nmd.2006.09.016 and others).

3. Part 1.5. It should be noted that female mdx mice show more pronounced signs of cardiac pathology, which is used in a number of groups.

4. It should be briefly noted that there are nematode models as well as dystrophin-deficient pigs.

5. Part 2.5. I also suggest noting changes in the level of creatine kinase (CK) MB fraction since its found primarily in heart muscle cells.

6. Part 3.2. I recommend noting not only the calcium-dependent activation of proteases, but also phospholipases, which contributes to the degradation of membrane phospholipids.

7. Line 711. This publication shows data from cardiac mitochondria, not skeletal muscle. In contrast, calcium uniport is suppressed in skeletal muscle, apparently due to an increase in the MCUb subunit.

8. Part 3.3. The hypothesis that the ADP/ATP antiporter (ANT) is involved in MPT opening is also discussed (including the mdx model of DMD). It appears that both ANT and ATP synthase components are involved in the formation of the pore channel, this is still an open question. Interestingly, recent data from Molkentin's group (albeit in sarcoglycan-deficient mice) also confirm the involvement of ANT in this phenomenon.

9. Lines 764 and 1087. I think there is a mistake here. This publication notes that glucocorticoids increase the sensitivity of skeletal muscle mitochondria to MPT pore opening.

10. In Part 3.3, the authors should note recent studies of the effects of alisporivir in mdx mice, showing possible suppression of mitochondrial biogenesis, dynamics and mitophagy in both muscle and heart, as well as the modifying effect of this agent on cardiac function, which is especially important in relation to the topic of the review.

11. Part 6.1. The authors should also note recent data on Dwarf Open Reading Frame (DWORF) gene therapy, which improved SERCA activity and mitigated the development of cardiomyopathy in mdx mice.

12. I did not find it in the review, but I recommend that the authors highlight data on changes in mitochondrial potassium homeostasis, which has been shown at least in mdx mice, this applies to both the heart and skeletal muscles. In both cases, improving potassium transport parameters using potassium channel modulators (so far I am aware of uridine and the benzimidazole derivative NS1619) also mitigates the development of cardiac pathology. This is a new area that requires attention.

13. Also as a recommendation, the authors should consider in more detail the issue of mitochondrial biogenesis, which is impaired in DMD and may be a target for therapy.

Author Response

REVIEWER 3

GENERAL COMMENT:

This is a very high-quality review demonstrating the various manifestations of cardiac pathology in Duchenne muscular dystrophy. The review focuses on mitochondrial dysfunction characteristic of this hereditary pathology. In general, the authors covered the problem well and emphasized further directions for research. However, I had several recommendations and comments regarding its content.

Minor:

  1. I recommend adding a table that summarizes data on the various manifestations of cardiac pathology in DMD (primarily mitochondrial, which is the subject of this review). The table should contain information about the object (human or model animal) demonstrating these manifestations.

Response: Given that several sections have been omitted or largely condensed (including the chamber-specific cardiomyopathy section that examined differences between human and pre-clinical models), and mitochondria-specific data is somewhat limited, we contest that adding a table would be redundant with information that is accessible in-text.

  1. Lines 144-147. It should be noted that a number of studies show the absence of a positive effect of Poloxamer 188 on the development of DMD (like doi: 10.1016/j.nmd.2006.09.016 and others).

Response: We added several additional references (including the one proposed by the Reviewer) between lines 160-166.

  1. Part 1.5. It should be noted that female mdx mice show more pronounced signs of cardiac pathology, which is used in a number of groups.

Response: Additional data and references have been included to highlight this unique phenomenon. Refer to lines 94-100.

  1. It should be briefly noted that there are nematode models as well as dystrophin-deficient pigs.

Response: An entire section has been added on the dystrophin-deficient pig and nematode models. See lines 260-268 (dystrophin-deficient pig) and 269-278 (nematode model). 

  1. Part 2.5. I also suggest noting changes in the level of creatine kinase (CK) MB fraction since its found primarily in heart muscle cells.

Response: CK-MB is found to be a hemodynamic biomarker in the final paragraph of Section 2.05, however, it is defined as a weak biomarker. As stated between lines 571-574, since CK-MB is present in both skeletal and cardiac muscle types, this protein is not a good marker for cardiac involvement in DMD patients as it can also be indicative of damage to other muscles in the body.

  1. Part 3.2. I recommend noting not only the calcium-dependent activation of proteases, but also phospholipases, which contributes to the degradation of membrane phospholipids.

Response: The data regarding phospholipase involvement in dystrophin-deficient cardiomyopathy is limited, however, available data on dystrophic myofibers has been newly incorporated. See lines 840-843.

  1. Line 711. This publication shows data from cardiac mitochondria, not skeletal muscle. In contrast, calcium uniport is suppressed in skeletal muscle, apparently due to an increase in the MCUb subunit.

Response: This error has been updated (skeletal muscle mitochondria replaced with cardiac mitochondria) and validated with the appropriate reference (Dubinin et al. 2020b). See lines 863-882.

  1. Part 3.3. The hypothesis that the ADP/ATP antiporter (ANT) is involved in MPT opening is also discussed (including the mdx model of DMD). It appears that both ANT and ATP synthase components are involved in the formation of the pore channel, this is still an open question. Interestingly, recent data from Molkentin's group (albeit in sarcoglycan-deficient mice) also confirm the involvement of ANT in this phenomenon.

Response: An additional sentence has been added to Section 3.03 (Mitochondrial permeability transition pore and apoptosis) to highlight data from sarcoglycan-deficient mice discussing the involvement of ANT with PTP. See lines 908-910.

  1. Lines 764 and 1087. I think there is a mistake here. This publication notes that glucocorticoids increase the sensitivity of skeletal muscle mitochondria to MPT pore opening.

Response: This sentence was removed when the extensive section on glucocorticoid therapies was condensed at the recommendation of the Reviewers.

  1. In Part 3.3, the authors should note recent studies of the effects of alisporivir in mdx mice, showing possible suppression of mitochondrial biogenesis, dynamics and mitophagy in both muscle and heart, as well as the modifying effect of this agent on cardiac function, which is especially important in relation to the topic of the review.

Response: An entire section on alisporivir’s role within mitochondrial biogenesis has been added to Section 3.04. See lines 973-1014.

  1. Part 6.1. The authors should also note recent data on Dwarf Open Reading Frame (DWORF) gene therapy, which improved SERCA activity and mitigated the development of cardiomyopathy in mdx mice.

Response: Given that we have largely condensed or omitted data from non-mitochondrial-targeting therapeutics, we have opted to not add additional gene therapy data by the recommendation of other Reviewers.

  1. I did not find it in the review, but I recommend that the authors highlight data on changes in mitochondrial potassium homeostasis, which has been shown at least in mdx mice, this applies to both the heart and skeletal muscles. In both cases, improving potassium transport parameters using potassium channel modulators (so far I am aware of uridine and the benzimidazole derivative NS1619) also mitigates the development of cardiac pathology. This is a new area that requires attention.

Response: We have added additional data (including the specific data we believe was requested by the Reviewer) between lines 863-882. This data has been strategically incorporated into Section 3.02 (Calcium handling dysregulation and cell death), where other ion homeostasis data is also discussed.

  1. Also as a recommendation, the authors should consider in more detail the issue of mitochondrial biogenesis, which is impaired in DMD and may be a target for therapy.

Response: Refer to point #10 above. We have added an entire new section on mitochondrial biogenesis as a potential therapeutic in DMD. See lines 973-1014.

Round 2

Reviewer 2 Report

Comments and Suggestions for Authors

Overall the authors were responsive to previous comments/suggestions. Some sections were reorganized or condensed and key points were clarified. 

I personally think that the manuscript still contains sections that are unnecessarily long,  given the purpose of the review. Section 2 in particular reviews clinical approaches to assess cardiac structure/function in patients with a systematic discussion of limitations inherent to each approach. That being said, the information conveyed is accurate and informative

Author Response

Comments: 

Overall the authors were responsive to previous comments/suggestions. Some sections were reorganized or condensed and key points were clarified. 

I personally think that the manuscript still contains sections that are unnecessarily long,  given the purpose of the review. Section 2 in particular reviews clinical approaches to assess cardiac structure/function in patients with a systematic discussion of limitations inherent to each approach. That being said, the information conveyed is accurate and informative.

Response: The authors appreciate the constructive criticism provided by the Reviewer and agree that this feedback has positively impacted the manuscript. The authors feel that Section 2 is necessary to convey current standard of care for DMD, prior to discussing mitochondrial therapeutics as an alternative therapeutic intervention. Given that the Reviewer agrees the information is accurate and informative, further streamlining it may create confusion for readers. The authors will keep these sections unaltered with the intentions that this manuscript benefits from a holistic understanding of the disease.

Reviewer 3 Report

Comments and Suggestions for Authors

The authors responded adequately to my comments.

Author Response

Comments: The authors responded adequately to my comments.

Response: The authors appreciate the feedback provided by the Reviewer and agree that the manuscript has positively benefited from the suggestions.